# Reliable crystal structure predictions from first principles

Rahul Nikhar [1] & Krzysztof Szalewicz [1 ✉]

An inexpensive and reliable method for molecular crystal structure predictions (CSPs) has been developed. The new CSP protocol starts from a two-dimensional graph of crystal's monomer(s) and utilizes no experimental information. Using results of quantum mechanical calculations for molecular dimers, an accurate two-body, rigid-monomer ab initio-based force field (aiFF) for the crystal is developed. Since CSPs with aiFFs are essentially as expensive as with empirical FFs, tens of thousands of plausible polymorphs generated by the crystal packing procedures can be optimized. Here we show the robustness of this protocol which found the experimental crystal within the 20 most stable predicted polymorphs for each of the 15 investigated molecules. The ranking was further refined by performing periodic density-functional theory (DFT) plus dispersion correction (pDFT+D) calculations for these 20 top-ranked polymorphs, resulting in the experimental crystal ranked as number one for all the systems studied (and the second polymorph, if known, ranked in the top few). Alternatively, the polymorphs generated can be used to improve aiFFs, which also leads to rank one predictions. The proposed CSP protocol should result in aiFFs replacing empirical FFs in CSP research.

[1] Department of Physics and Astronomy, University of Delaware, Newark, DE 19716, USA. ✉email: szalewic@udel.edu

Properties of crystalline solids depend critically on the polymorphic form of a given substance and many crystals can exist in several such forms[1,2]. The knowledge of possible stable polymorphic forms of a crystal is of particular importance for pharmaceutical industry[3]. If a polymorph different from the one obtained in laboratories crystallizes during manufacturing of a drug, it will have different physicochemical properties and may lead to undesirable therapeutic effects, two examples are ritonavir[4,5] and rotigotine[6–8]. Thus, in the drug development process, one would like to know if the polymorph used is thermodynamically the most stable form in ambient conditions. In defense industry, developments of energetic materials are costly and highly dangerous[9,10] and a priori knowledge of crystal structure of notional materials would allow accelerated screening of such materials. Also semiconductor industry can benefit from such knowledge[11]. CSP methods answer these needs by finding a set of most stable crystalline polymorphs of a given molecule starting from its two-dimensional diagram and not using any experimental information specific for this molecule.

Reliable CSPs for molecular crystals starting from the knowledge of only two-dimensional diagrams of monomer(s) were nearly impossible for a long time. In 1988, Maddox[12] described failure of CSPs as a continuing scandal in the physical sciences and stated that in general even simplest crystalline solids posed great challenge. In mid 1990s, Gavezzotti[13] asked the fundamental question: 'Are crystal structures predictable?', and his answer was 'No'. In response to this criticism, the Cambridge Crystallographic Data Centre (CCDC) conducted a series of "blind" tests[14–19] by providing only two-dimensional diagrams of monomers of crystals that have been measured but not published and asking research groups to submit their predictions, with the results of the first test published in 2000. While the field has advanced significantly since the first test, the results of the last, 6th test[19] are still not completely satisfactory. The participating groups achieved the success rate between 13% and 57% (not including polymorphs C and E of system XXIII), where success means that the experimental polymorph was found among polymorphs on two lists containing 100 polymorphs each.

One should remark here that predictions of crystals structures are actually a difficult problem for physical science, opposite to what Maddox[12] implied. The reason is the high dimensionality of the conformational and crystallographic space resulting in thousands of plausible polymorphs produced by sampling of this space within a relatively narrow window of lattice energies and densities. The energetic distances between consecutive polymorphs ordered by lattice energy are of the order of 1 kJ/mol at the low-energy end, which requires accuracies nearly impossible to achieve by empirical FFs. Also, for experimentally observed polymorphs, the differences between their computed lattice energies are of the same order[20].

While there are several variants of CSPs, including a recent use of deep neural networks[21], the majority of groups participating in the 6th blind test used some form of FFs, mostly of empirical character. The most successful CSP protocol consisting of polymorph-space sampling plus lattice-energy minimization has been developed by Neumann et al.[22,23]. This protocol uses a tailor-made FF which is obtained by refining parameters of an empirical FF to reproduce as close as possible pDFT+D lattice energies (and their derivatives). The initial polymorphs for pDFT+D calculations are obtained using the empirical FF. The method is included in the commercial software package GRACE (Generation Ranking and Characterization Engine)[24], but some of its details are not available. Recent reviews of the field of CSPs can be found in refs. [25–31].

In the present work, a CSP protocol is proposed based entirely on first principles, i.e., not utilizing any experimental information. Since the main characteristic of this method is the use of aiFFs, we will refer to it as the CSP(aiFF) protocol. This protocol consists of several stages shown in Fig. 1. While aiFFs have been used in CSPs for some time[19,32–34], such predictions were taking a long time (several months at the minimum), required huge amounts of human effort, and were possible for monomers with up to about 20 atoms. In the present work, four recent developments are combined to dramatically reduce costs and increase predictability of such CSPs: (a) The development of a very effective variant[35] of symmetry-adapted perturbation theory (SAPT)[36] for ab initio calculations of interaction energies; (b) The creation of autoPES[37,38]: an automatic, effective, and reliable method for generation of potential energy surfaces (PESs) with minimal human involvement; (c) Enabling the use such aiFFs in the lattice-energy minimization stage of CSPs, a part of the present work; (d) The application of pDFT+D for a final refinement of polymorph rankings. Stage 3 of Fig. 1 can produce even millions of polymorphs at low costs and past experience indicates that the experimentally relevant polymorphs are almost always among them. Thus, the essence of CSP protocols is to filter all relevant low lattice energy polymorphs out of this set. In the past few years, it has been demonstrated by several groups that pDFT+D geometry optimization of polymorphs places the experimental polymorph ranked within the top few, often as number one[19,39–41]. However, such calculations are so expensive that they can be afforded for only a hundred or so polymorphs. In contrast, if an FF is used in Stage 4, tens of thousands polymorphs can be optimized. This FF has to be sufficiently accurate not to miss any important polymorphs. Thus, both the ab initio method and the fit to interaction energies computed using this method must have sufficiently small uncertainties. In calculations of dimer interaction energies, the variant of SAPT used by us (see "Methods") is nearly as accurate[35,42,43] as the coupled cluster method with single, double, and noniterative triple excitations, CCSD(T), the "gold-standard" method of electronic structure theory, but is significantly less expensive. To prevent loss of accuracy due to fitting, the form of the fitting function has to be significantly more involved than those of empirical FFs that are typically built from Lennard-Jones plus Coulomb potentials, see "Methods". The extended form can fit ab initio data with uncertainties of about 1 kJ/mol, which we will show to be sufficient for reliable CSPs. Such form has never been used in lattice energy minimizations and we had to modify CSP software to make it possible. Finally, to make Stage 2 affordable, the number of ab initio grid points needed to fit an aiFF has to be reasonably small. The autoPES method[37,38] reduces this number by two orders of magnitude compared to typical surface-fitting approaches, reducing in this way the development costs by the same ratio. It also reduces amount of human involvement almost to zero as the whole process is completely automated. We show below that the proposed protocol found the experimental crystal ranked as number one for all 15 molecules studied (and the second polymorph, if known, ranked in the top few).

## Results and discussion

**Performance of CSP(aiFF) protocol**. To asses the performance of our method, we carried out CSPs for 15 molecules including several systems from the CCDC blind tests[14–19] (denoted by roman numerals), as well as for methanol, benzene, nitromethane, 5,5′-dinitro-2H,2′H-3,3′-bi-1,2,4-triazole (DNBT), 1-3-5-trinitrobenzene (TNB), deferiprone, and fluorouracil. The molecular graphs are shown in Supplementary Fig. 1. The results are summarized in Table 1. An extended version of this table is available in Supplementary Table 1.

The CSP(aiFF) protocol ranked the experimental polymorph as number 1 in 5 cases, as number 2–6 in 7 cases, and as numbers 9, 9, and 16. We have also included a second experimentally

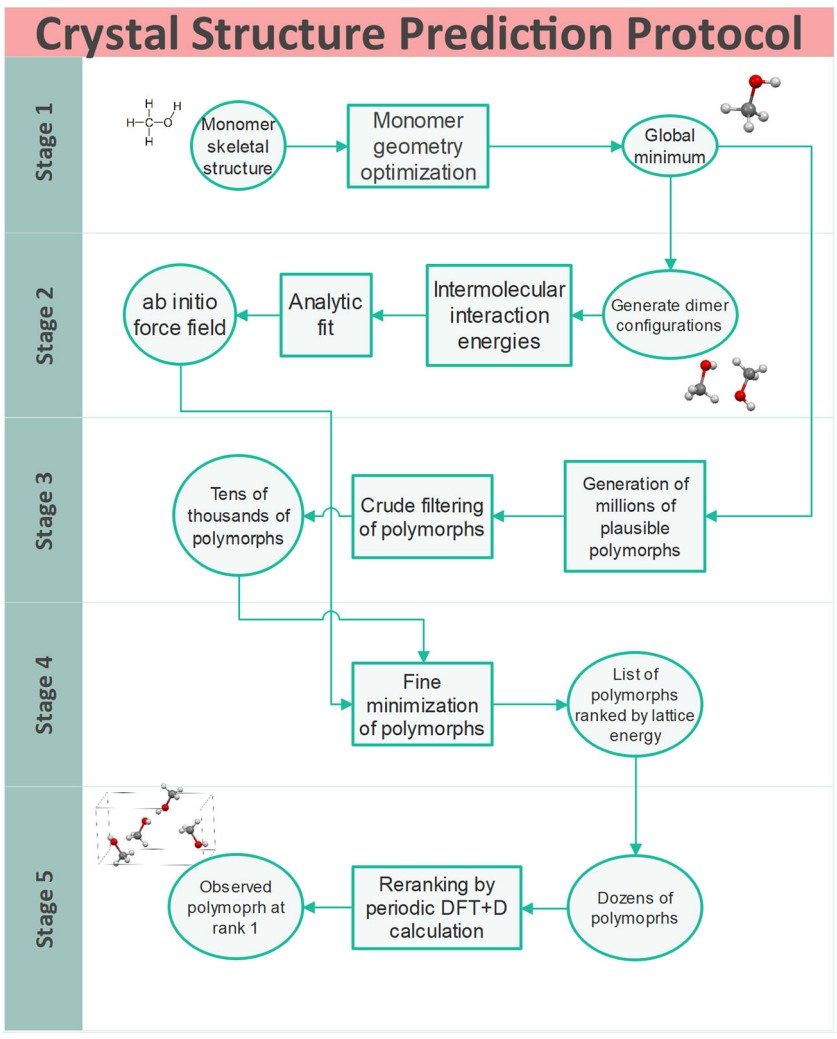

**Fig. 1 Overview of aiFF-based CSP protocol.** Stage 1: monomer energy minimization to find the equilibrium geometry. Stage 2: ab initio calculations of dimer intermolecular interaction energies followed by fitting an analytic form of aiFF to these data. Stage 3: generation of millions of plausible packing arrangements of polymorphs by sampling different space groups, orientations of monomers, and unit cell parameters, followed by a reduction of this set to tens of thousands of polymorphs using density criteria or crude lattice energy minimizations with simple FFs. Stage 4: fine minimization with aiFFs for all polymorphs in the reduced set. Stage 5: refinement of the ranking via pDFT+D calculations on a couple dozen top-ranked polymorphs from Stage 4.

identified polymorph in the cases of system I, benzene, and deferiprone, denoted as "Poly2" in Table 1, and these are ranked as numbers 8, 4, and 8, respectively. After pDFT+D calculations on top-ranked 20 polymorphs of each crystal, without any further geometry optimization, an experimental crystal became ranked as number 1 in each case. For deferiprone, it was Poly2 that became the rank 1 polymorph, while Poly1 remained at rank 2. For system I and benzene, Poly2 changed rank from 8 to 2 and from 4 to 3, respectively. $RMSD_{20}$'s between the calculated and experimental crystals vary between 0.09 and 0.67 Å, below the CCDC threshold of 0.8 Å. Also densities and cell parameters, shown in Supplementary Table 1, agree very closely. Supplementary Fig. 2 displays the percent deviations between the calculated and experimental lattice parameters. The average errors for the cell parameters $a$, $b$, $c$, and $\beta$ amount to 4.3%, 2.6%, 4.3%, and 2.4%, respectively. Such level of predictivity is unprecedented for a completely first-principles CSP protocol. The overlaps of the experimental polymorphs with the closest calculated ones are shown in Fig. 2. This figure allows intuitive appreciation how close these structures are. This exceptional performance of CSP(aiFF) has been achieved despite the investigated systems exhibiting typical difficulties due to closeness of polymorphs'

lattice energies and despite using rigid-monomer approximation. The lattice energy vs. density landscapes from the aiFF minimizations for systems IV and XXII are shown in Supplementary Fig. 3. Analogous graphs for the other systems look similar. The lowest-energy 100 polymorphs spread the range of about 5 kJ/mol for systems I, XII, XIII, benzene, and nitromethane, about 10 kJ/mol for systems II, IV, VIII, XVI, XXII, methanol, TNB, deferiprone, and fluorouracil, and about 20 kJ/mol for DNBT. At the low-energy end, the energy differences between consecutive polymorphs are less than 1 kJ/mol, i.e., comparable to the RMSEs of the fits over all dimer configurations with negative interaction energies, shown in Table 1.

**Performance of a simplified aiFF form**. The use of the extended functional form of aiFFs in the lattice energy minimizations instead of the simpler exp-6-1 form (not including a polynomial in front of exponential, damping functions, etc., see "Methods") used in some empirical FFs leads to enormous improvements in rankings. To quantify such improvements, we performed lattice energy minimizations with the exp-6-1 form of aiFFs, fitted using the same level of theory as in the case of the extended form, for systems I, II, IV, and XXII, achieving rankings of 138, 2231, 49,

and 60, respectively, while the rankings of the extended aiFF form for these systems are 1 or 2, see Table 1. The main reason for this improvement is that RMSEs for negative interaction energies are from 2.3 to 5.3 times smaller in the latter case (these ratios are correlated with the number of fit parameters: e.g., 30 and 270 for the exp-6-1 and the extended form, respectively, in the case of system IV).

**Performance of an empirical FF.** In order to quantify better the predictive power of our approach, calculations analogous to those described above have been performed with an empirical FF. We have chosen the W99 FF[44] with point charges computed by us using the CHELPG method[45]. For the 18 experimental polymorphs considered, the W99+charges FF found 33% of them at rank 10 or better, while the analogous result for aiFF (without the pDFT+D step) is 94%. This amounts to a qualitative difference for technological applications. For more details on CSPs with the W99+charges FF, see Supplementary Tables 1 and 2.

**Alternative CSP(aiFF) protocol.** One may ask why pDFT+D calculations are needed to improve the rankings, while several comparisons on benchmark interaction energies, see, e.g., refs. [42,43], show that SAPT(DFT) is nearly as accurate as CCSD(T) and more accurate than DFT+D methods. The main reason is that what is used in CSPs are aiFFs, and they include additional uncertainties due to fitting. Although the average fit error for negative interaction energies is only ~1 kJ/mol, errors may be larger at some configurations. If a configuration with a too negative interaction energy is important for a polymorph, this polymorph may become overly stable and therefore too highly ranked. Two other possible reasons, basis set size and neglect of many-body effects in CSP(aiFF), are discussed in Supplementary Information and found unlikely to be a reason. To improve the predictions from Stage 4, we have developed an alternative version of our method, alt-CSP(aiFF). After executing the CSP(aiFF) protocol less the pDFT+D stage, the geometries of 20 polymorphs with the lowest lattice energies are examined and consecutive nearest neighbor dimers identified. Then SAPT calculations are performed for these dimers, the aiFF is refitted, and lattice minimizations for the 20 polymorphs are performed with the new aiFF. This procedure is iterated until the energies of

the 5x5x5 clusters extracted from each polymorph computed in two ways: just from the aiFF and in a hybrid way, replacing the aiFF interaction energies by the available SAPT ones, are the same to within some threshold. We have applied alt-CSP(aiFF) to two of the worst ranking crystals from Table 1: system XVI (rank 16) and fluorouracil (rank 9). In each case, alt-CSP(aiFF) resulted in the experimental polymorph at rank 1, while $RMSD_{20}$ was reduced from 0.29 to 0.15 Å and from 0.61 to 0.42 Å, respectively. Thus, alt-CSP(aiFF) can be used without the pDFT+D stage. However, the additional ab initio calculations are about as expensive as the pDFT+D ones, so there is no gain in terms of efficiency.

**Table 1 CSPs from SAPT(DFT)-based aiFFs minimizations followed by pDFT+D fixed-geometry calculations.**

| System | SG | Rank | $RMSD_{20}$ | RMSE |
|---|---|---|---|---|
| I$_{Poly1}$ | $P2_1/c$ | 2/1 | 0.09 | 0.6 |
| I$_{Poly2}$ | $Pbca$ | 8/2 | 0.32 | 0.6 |
| II | $P2_1/n$ | 1/1 | 0.59 | 1.3 |
| IV | $P2_1/c$ | 2/1 | 0.24 | 0.63 |
| VIII | $C2/c$ | 4/1 | 0.28 | 1.1 |
| XII | $Pbca$ | 9/1 | 0.53 | 0.84 |
| XIII | $P2_1/c$ | 4/1 | 0.45 | 1.1 |
| XVI | $Pbca$ | 16/1 | 0.29 | 1.0 |
| XXII | $P2_1/n$ | 1/1 | 0.15 | 1.4 |
| Methanol | $P2_12_12_1$ | 6/1 | 0.4 | 0.92 |
| Benzene$_{Poly1}$ | $Pbca$ | 1/1 | 0.16 | 0.59 |
| Benzene$_{Poly2}$ | $P2_1/c$ | 4/3 | 0.4 | 0.59 |
| Nitromethane | $P2_12_12_1$ | 1/1 | 0.27 | 0.74 |
| DNBT | $P2_1/c$ | 1/1 | 0.58 | 1.56 |
| TNB | $P2_1/c$ | 3/1 | 0.67 | 1.28 |
| Deferiprone$_{Poly1}$ | $Pbca$ | 2/2 | 0.28 | 0.71 |
| Deferiprone$_{Poly2}$ | $P2_1/c$ | 8/1 | 0.24 | 0.71 |
| Fluorouracil | $P2_1/c$ | 9/1 | 0.61 | 1.06 |

SG: predicted space group of the crystal (SG is the same for experimental and predicted polymorphs); Rank: rank of the experimental polymorph after minimizations and after pDFT+D calculations; $RMSD_{20}$: root mean square deviation (in Å) between the experimental crystal and the calculated polymorph for 20 overlapping molecules (heavy atoms only); RMSE: root mean square error (in kJ/mol) of the fit for negative interaction energies.

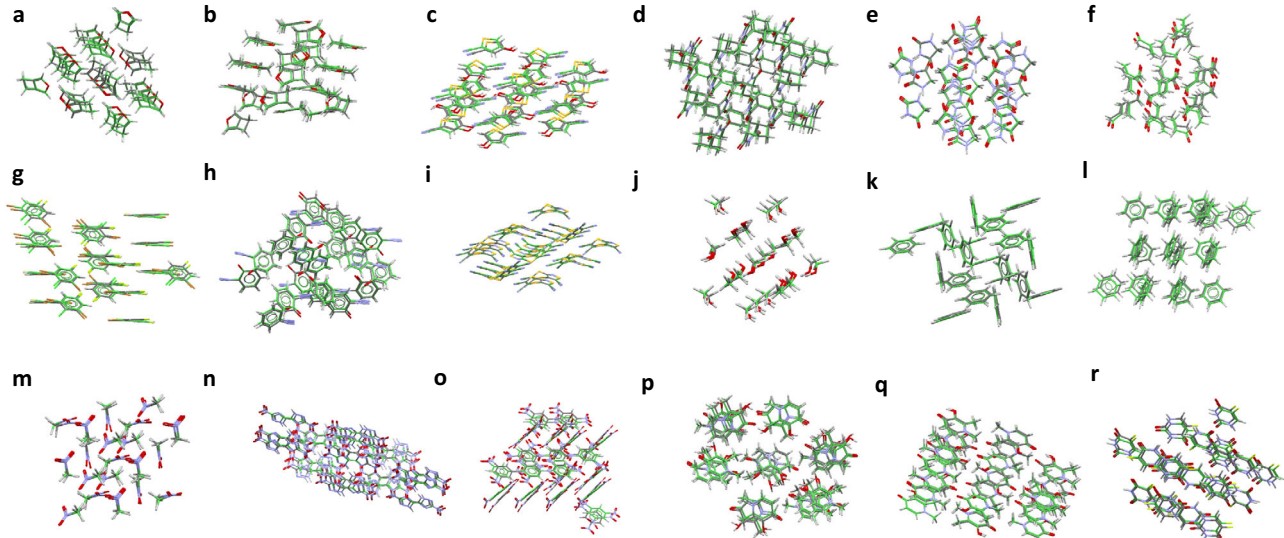

**Fig. 2 Overlaps of crystal structures.** Overlap of the experimental crystal structure (element-specific colors) with the closest calculated crystal structure (green) using SAPT(DFT)-based aiFFs for systems: **a** and **b** I, **c** II, **d** IV, **e** VIII, **f** XII, **g** XIII, **h** XVI, **i** XXII, **j** methanol, **k** and **l** benzene, **m** nitromethane, **n** DNBT, **o** TNB, **p** and **q** Deferiprone, **r** Fluorouracil.

**Cost comparisons.** The method proposed not only is highly reliable, as shown above, but also is very efficient compared to alternative ways of combining FF-based CSPs with pDFT+D calculations. To demonstrate this, we show in Fig. 3 the costs of three possible CSP strategies in terms of single-core wall times on the example of system I. Note that this type of calculations are typically performed on hundreds of cores, so the actual wall time is just a couple hours for Strategy 1, the approach proposed here. The majority of time for Strategy 1, 7 core-days, is spent for the development of an aiFF and most of this time is used to compute SAPT(DFT) interaction energies for 706 dimer configurations, with very little time spent on fitting these energies. The next stage, the packing and minimization (PACK+MIN) of hundreds of thousands of crystals, requires only less than a third of a day. The final stage, pDFT+D calculations for the top 20 polymorphs at aiFF geometries, requires approximately one day. Hypothetical Strategy 2 differs from Strategy 1 by the use of an empirical FF in the PACK+MIN stage and by performing pDFT+D calculations for 100 polymorphs with reoptimization of geometries (this number of polymorphs was chosen as a trade-off between success rate and computational costs). The time required for the latter stage would be 70 core-days, so Strategy 2 is about an order of magnitude more expensive than Strategy 1. Moreover, if the W99+charges FF were used, the success rate of Strategy 2 on the set of 18 polymorphs examined here would be 72% (see Supplementary Table 2), while the success rate of Strategy 1 is 100% already with 16 top-ranked polymorphs. All the PACK+MIN bars appear to be of about the same height for aiFF and for the empirical FF. This is because the calculation of the lattice energy is only about two times more expensive in the former case. Hypothetical Strategy 3 performs pDFT+D calculations with geometry optimization for all 25,500 polymorphs produced by PACK+MIN. This strategy would have a very high reliability (since practice indicates that the experimental polymorphs are almost always included in such a large pool of candidate structures), but it would be extremely costly, 49 single-core years, and hence not practical (although possible if a few thousands cores were used). With the use of an empirical FF, one can set the number of polymorphs included in the pDFT+D stage anywhere between 100 and 25,000, systematically increasing costs and reliability relative to Strategy 2. However, with W99+charges and our set of polymorphs, the success rate would remain at 72% until the number of polymorphs is at least 589. For Strategies 2 and 3, the PACK+MIN stage can be replaced by any other protocol producing the required number of candidate polymorphs, with insignificant effects on the total timings.

**Neglected effects.** Since aiFFs are sums of two-body interactions, they neglect the many-body effects mentioned earlier and discussed in Supplementary Information. While we show that these effects are not critical in CSPs for the crystals considered here, they may be significant for some other crystals[46–48]. The most important many-body effect, the many-body polarization, can be accounted for using polarizable aiFFs that can be developed using autoPES, but are not yet implemented in our CSP codes. In Supplementary Information, we also explain why the relatively small basis set that we used is adequate for CSPs. A much more important neglected effect is flexibility of monomers. Although the monomers considered by us were assumed to be rigid, the proposed CSP(aiFF) protocol can be applied to monomers with soft degrees of freedom. Such monomers may be significantly deformed in crystals compared to their equilibrium structures in gas phase. The recent version of autoPES[38] has the capability of computing interaction energies accounting for all or selected intramonomer degrees of freedom and most CSP codes can

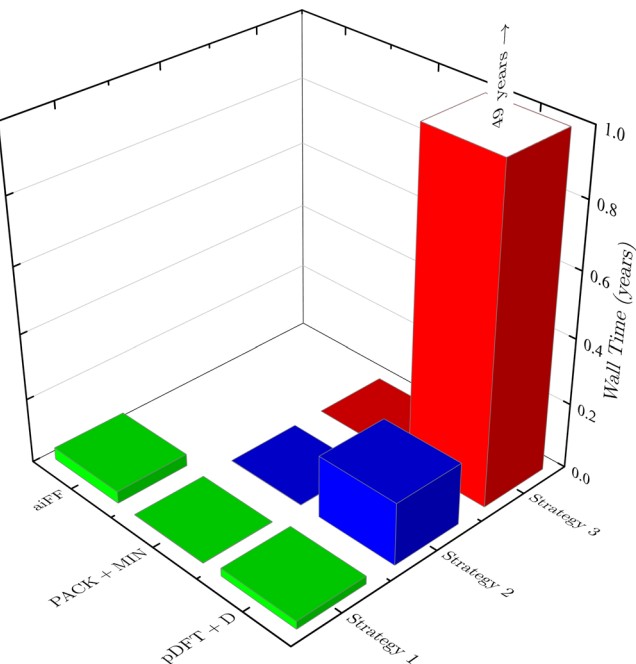

**Fig. 3 Computational cost of the considered CSP protocols.** Total wall times required for system I CSPs on a single core of the Intel E5-2670 processor using different strategies. Rows "aiFF", "PACK+MIN", and "pDFT+D" denote times of an aiFF development, packing and minimization, and of periodic DFT+D calculations.

perform packing and minimization including all degrees of freedom, therefore such predictions can be made still completely from first principles. However, costs of such calculations increase steeply with the total number of degrees of freedom. One way around this problem is to assume separation of inter- and intramonomer degrees of freedom in Stage 2, as it has been done in all biomolecular FFs and in all FFs used in flexible-monomers CSPs. Since our aiFFs depend only of separations between atoms of different monomers, interaction energies can be computed for arbitrary monomer configurations. Such "flexibilized" intermonomer FF can replace the intermonomer component of current empirical FFs, while the intramonomer component can be kept unchanged. One can expect that such a replacement should lead to improved predictions in flexible-monomer CSPs.

Other effects neglected by the present version of CSP(aiFF) are thermal and entropic ones, as the results presented by us correspond to 0 K temperature. For some crystals, these effects can change the rankings of polymorphs, as pointed out by Brandenburg and Grimme[39] and recently investigated extensively by Hoja et al.[41]. The thermal and entropic effects can be routinely computed using pDFT+D, although such calculations are several times more expensive than pDFT+D calculations with static geometries. As a test, we have computed both effects for the 5 lowest lattice energy polymorphs of system XXII, leading to no change of rankings.

**Concluding remarks.** The first-principles CSP(aiFF) method developed here was applied to crystals of 15 rigid molecules with 18 known experimental polymorphs. When aiFFs are applied in CSPs for crystals of these molecules, 17 or 94% the polymorphs are ranked in the range 1–10, while the remaining one has rank 16. For comparison, analogous CSPs with the empirical W99+charges FF ranks only 33% of polymorphs in the range 1–10, 3 experimental polymorphs are not found within 568 or more generated ones, and for two molecules predictions were not

possible due to missing atom types. The ability of CSP(aiFF) to minimize tens of thousands polymorphs is its key advantage over alternative approaches which have to use low-accuracy methods at this stage, often erroneously leading to discarding of correct structures. Upon a subsequent reranking of the top 20 polymorphs with pDFT+D calculations at fixed aiFF geometries, for all 15 molecules an experimental polymorph became ranked as number 1, while the second polymorphs became ranked as numbers 2, 2, and 3. The pDFT+D step can be omitted if aiFFs are iteratively improved by performing ab initio calculations on dimers extracted from crystals predicted with the previous iteration of an aiFF [the alt-CSP(aiFF) protocol]. The proposed CSP protocol not only shows ultimate predictive power for the systems tested, but is also inexpensive compared to other highly predictive approaches. On about a hundred cores, complete predictions for any of the systems investigated here take less than a day, including the aiFF generation. The CSP(aiFF) protocol requires a minimal human involvement, consisting only of input preparation for autoPES, UPACK, and pDFT+D calculations, and includes only free software with open source codes. Limitations of the current implementation of the CSP(aiFF) methodology have been discussed, in particular the neglect of many-body interactions and the rigid-monomer approximation. Although the test set included only homogeneous crystals, there are no reasons to doubt that the method will work equally well for cocrystals including salts since the quality of aiFFs does not depend on dimers being homogeneous or heterogeneous (of course, for two-component cocrystals, three PESs have to be developed). Also, while the largest of the test molecules included 22 atoms, the method should apply equally well to larger molecules since the relative accuracy of SAPT(DFT) does not change with system size[35]. Of course, calculations will be more expensive as the size increases, but molecules with about 100 atoms are within reach. The effectiveness of the proposed CSP protocol is due to the use of the SAPT(DFT) method which is computationally efficient relative to other accurate electronic structure methods and due to the use of the autoPES method for fitting aiFFs since this method not only cuts the costs of such fits by orders of magnitude, but also reduces human effort of this most difficult to automate step almost to zero. An important element of the CSP(aiFF) protocol is that it replaces simple potential forms used in all earlier CSP protocols by an extended form capable of fitting ab initio interaction energies with significantly decreased uncertainties. An advantage of the proposed protocol is that it constitutes a complete first-principles procedure for investigating crystal structures and properties. Such a protocol should work equally well for any type of monomer, in contrast to the protocols using empirical FFs, which are expected to work well only for systems similar to those used in fitting such FFs. We believe that the overall effect of the proposed CSP protocol will be that the field of CSPs will move from the use of empirical FFs to aiFFs. This should increase reliability of predictions and therefore, while CSPs have played so far at the best advisory role in technology developments, they may become a leading element in developments of novel crystalline materials. More generally, aiFFs can be used in several types of computational material design.

## Methods

**Monomer geometry minimization**. In Stage 1, monomer geometries were optimized using ORCA[49,50] with the PBE[51] functional and D3 correction[52] in the aug-cc-pVTZ[53] basis set.

**Ab initio calculations of interaction energies**. To make the CSP(aiFF) protocol practical, aiFFs have to be constructed in Stage 2 at reasonably low costs, but at the same time with small uncertainties, for monomers with dozens of atoms. This requires first that the ab initio method used to compute intermolecular interaction energies is inexpensive and accurate. It appears that the best current choice for such calculations is SAPT[36,54], an ab initio method that computes interaction energies directly, starting from isolated monomers and imposing the correct electron

permutational symmetry. We applied the SAPT variant based on DFT, SAPT(DFT)[55,56], see ref. [35] for a recent review of this method. SAPT(DFT) and CCSD(T) calculations scale as $\mathcal{O}(n^5)$ and $\mathcal{O}(n^7)$ with system size, respectively, where $n$ is the number of electrons, and for dimers with a couple dozens of atoms, SAPT(DFT) calculations are about two orders of magnitude less expensive than CCSD(T) calculations. The recently developed new SAPT(DFT) algorithms and effective computer codes[35,42] can be used to compute thousands of grid points for dimers with ~100-atom monomers using reasonable computer resources and being able to achieve this in a few days if a sufficient number of computer cores are available.

The details of calculations of SAPT(DFT)[55–58] first- and second-order interaction energies are as follows. We used the density-fitting version[35,59,60] in the SAPT2020[61] codes interfaced with the ORCA package[49,50] for calculations on monomers. The PBE[51] functional was used in DFT calculations applying the gradient-regulated asymptotic correction (GRAC)[62,63]. The aug-cc-pVDZ[53] basis set plus a set of 3s3p2d2f midbond functions (default of autoPES) was used in the monomer-centered plus basis set (MC⁺BS) format[64]. The terms accounting for higher-order induction and exchange-induction effects, denoted as $\delta E_{\mathrm{int,resp}}^{\mathrm{HF}}$ and obtained as a difference between Hartree–Fock (HF) interaction energies and the sum of appropriate SAPT(HF) first- and second-order corrections in their response (resp) versions, was included for all systems except system XIII, benzene, DNBT, and TNB. We use a short-hand notation for SAPT interaction energy components: "indx" is the sum of the second-order induction and exchange-induction components, as well as of the $\delta E_{\mathrm{int,resp}}^{\mathrm{HF}}$ contribution, "dispx" is the sum of the dispersion and exchange-dispersion components, "elst" is the electrostatic component, and "exch" is the first-order exchange component. Relative importance of attractive components is illustrated in Supplementary Fig. 4.

**Generation of aiFFs**. In all past CSPs, only simple FFs have been used at the lattice-energy minimization stage. The two most often used forms are the Lennard-Jones 12-6-1 potential: $A_{12}/r^{12} - C_6/r^6 + q_a q_b/r$, and the Buckingham exp-6-1 potential: $Ae^{-\beta r} - C_6/r^6 + q_a q_b/r$, where $r$ is an atom-atom distance and $A_{12}$, $A$, $\beta$, $C_6$, $q_a$, and $q_b$ are adjustable parameters. SAPT(DFT)-based aiFFs have been used in CSPs, but always with the exp-6-1 potential form in the minimization stage. This form is not pliable enough to fit well ab initio data, leading to uncertainties of a few kJ/mol, too large for reliable CSPs. In contrast, the extended form used by us in the CSP(aiFF) protocol can fit ab initio data with uncertainties of about 1 kJ/mol, which we show to be sufficient for reliable CSPs. This functional form is[37,38]

$$V = \sum_{a \in A, b \in B} \left\{ \left[ 1 + \sum_{i=1,2} a_i^{ab}(r_{ab})^i \right] e^{\alpha^{ab} - \beta^{ab} r_{ab}} + \frac{A_{12}^{ab}}{(r_{ab})^{12}} \right. \\ \left. - \sum_{n=6,8} f_n(\delta_n^{ab}, r_{ab}) \frac{C_n^{ab}}{(r_{ab})^n} + f_1(\delta_1^{ab}, r_{ab}) \frac{q_a q_b}{r_{ab}} \right\} \quad (1)$$

where $a$ ($b$) goes over the sets of atoms in monomer A (B), respectively, $\alpha^{ab}$, $\beta^{ab}$, $a_i^{ab}$, $A_{12}^{ab}$ are repulsion-energy parameters, $C_n^{ab}$ are long-range dispersion plus induction energy parameters, $q_x$, $x = a$, $b$, are atomic partial charges, $\delta_n^{ab}$ are damping parameters, and $f_n$ are the Tang-Toennies[65] damping functions: $f_n(\delta, r) = 1 - e^{-\delta r} \sum_{m=0}^{n} (\delta r)^m / m!$ Long-range interaction energies were computed using an ab initio-distributed approach. The damping parameters in the dispersion plus induction term were fitted separately to the sum of all close-range second-order components plus $\delta E_{\mathrm{int,resp}}^{\mathrm{HF}}$, while $\delta_1^{ab}$ were fitted to electrostatic energies. All PESs developed here are two-body, 6-dimensional PESs, i.e., assume rigid monomers. The aiFFs were constructed as sums of these two-body PESs. One should add that the extended form of FFs given by Eq. (1) has been used in some published CSPs, but only in molecular dynamics (MD) simulations that can replace the pDFT+D calculations of Stage 5. Note that MD calculations are about as expensive as pDFT+D ones and significantly more expensive than the minimizations of Stage 4. Graphs showing SAPT(DFT) interaction energy components and their fits as functions of the distance $R$ between the centers of mass of monomers are included in Supplementary Fig. 5. One can see in particular that the ab initio electrostatic energies are reproduced very well for $R$'s larger than the van der Waals minimum distance $R_{\mathrm{vdW}}$ despite using only damped charge-charge interactions, i.e., omitting higher multipolar terms. While the use of the latter terms in empirical FFs improves the predictions compared to the use of charges only[66,67], our results show that higher-rank multipoles are not needed if the electrostatic function includes damping and is fitted to ab initio electrostatic energies. The worsening of the agreement with ab initio values seen for $R < R_{\mathrm{vdW}}$ is inevitable and is due to the charge-overlap effects that are not proportional to inverse powers of $R$[36]. These effects are accounted for in the overall fit by the first term in Eq. (1). This is why the total fitted and ab initio interaction energies are in excellent agreement for all $R$.

**Crystal packing and lattice-energy minimization**. Since none of the available CSP packages is capable of using the form of aiFFs given by Eq. (1), we have modified two such packages: MOLPAK[68] and UPACK[69] to be applied in Stages 3 and 4. MOLPAK uses the concept of coordination geometry and by default searches in 26 space groups: $P1$, $P\bar{1}$, $P2$, $Pm$, $Pc$, $P2_1$, $P2/c$, $P2_1/m$, $P2/m$, $P2_1/c$, $Cc$, $C2$, $C2/c$, $Pnn2$, $Pba2$, $Pnc2$, $P22_1$, $Pmn2_1$, $Pma2$, $P2_12_12$, $P2_12_12_1$, $Pca2_1$, $Pna2_1$, $Pnma$, $Fdd2$, $Pbcn$, and $Pbca$. It generates polymorphs on a grid in three-dimensional

search space by systematically varying the orientation of the central molecule in steps of 10°. This generation is performed in all 51 coordination geometries. The packing in the unit cell is controlled by a simple repulsive $1/r^{12}$ interaction between atoms: the molecules are brought together until an energy threshold is reached. This step provides an initial set of 6859 angle combinations × 51 coordination geometries = 349,809 hypothetical polymorphs. From this set, 25,500 densest polymorphs, 500 from each coordination geometry, are minimized using the program WMIN[70]. The default functional form of the FF in WMIN is exp-6-1. We have modified this code to include FFs of the form of Eq. (1).

UPACK generates random crystal structures in 13 default space groups: $C2$, $C2/c$, $Cc$, $P1$, $P\bar{1}$, $P2_1$, $P2_1/c$, $P2_12_12_1$, $Pbca$, $Pc$, $Pbcn$, $Pca2_1$, and $Pna2_1$. It can use any 12-6-1 potential and we selected the OPLS-AA FF[71]. The packing stage is divided in UPACK into two steps. In the first step, only 500 reasonable structures per symmetry group are randomly generated in an unrestricted way and are then used to estimate cell dimensions. In the second step, the random generation is performed in a restricted coordinate space using this cell estimate. Most of the generated structures are immediately rejected using the criterion that atom-atom 12-6-1 interactions are not allowed to be larger than 2000 kJ/mol for any pair. Such generations plus energy criterion testings continue until 5000 polymorphs per symmetry group, i.e., the total of 65,000 polymorphs are found. This second step involves also a rough optimization of lattice energies. The resulting list is subjected to clustering[72] to remove duplicates. Clustering reduces the pool significantly. For example, for system XXII it is reduced to 13,014 polymorphs.

In Stage 4 of CSP(aiFF) realized with UPACK, all the polymorphs from the reduced set are minimized with tight thresholds. We have modified UPACK to be able to use FFs of the form of Eq. (1). We found that it is advantageous to perform Stage 4 first with the OPLS-AA FF, i.e., using the original UPACK path including clustering, and then minimize the reduced set using aiFF. The procedure was chosen not to save time, although it does result in minor savings, but to avoid minimizations ending up in "holes" of an FF, i.e., unphysical minima at very short intermonomer separations. By construction, 12-6-1 FFs do not have any holes, while exp-6-1 and our extended-form FFs almost always have holes (although behind about 100 kJ/mol barriers, one of constraints of the autoPES fitting). We found that aiFF minimizations starting from the OPLS-minimized structures almost never end up in holes. We could have easily avoided the use of OPLS by fitting a 12-6-1 FF to the ab initio data.

The two CSP packages modified by us produced almost identical predictions for cases where we used both. MOLPAK was used for systems I, II, XII, XXII, nitromethane, and benzene. UPACK was used for the remaining systems, as well as for system I, II, and XXII treated also by MOLPAK. For these three systems, rankings of the experimental crystal by the two packages were identical.

PLATON[73] was used for checking missed symmetries[74] and for space group transformations from non-standard setting to standard setting by assigning the target crystal the proper space group and cell parameters, leading to the data in Table 1. For example, for system II both MOLPAK and UPACK predicted the experimental crystal in $P2_1/c$ symmetry, and PLATON transformed it to $P2_1/n$ symmetry.

**pDFT+D calculations**. In Stage 5, periodic single-point DFT+D lattice energy calculations, i.e., without geometry optimizations, were performed for the 20 top-ranked polymorphs from aiFF minimizations using the PBE[51] functional with pseudopotentials[75] plus the D3 dispersion correction[52] with the Becke–Johnson (BJ) damping[76,77]. We used Quantum ESPRESSO (QE)[78,79] codes, with the plane-wave kinetic energy cutoffs of 340 and 3061 eV for the wave functions and charge densities, respectively.

The zero-point vibrational energy (ZPVE) and thermal effects were calculated within the harmonic approximation using Phonopy 2.8.1[80] and VASP 5.4.4[81–85] with the same DFT+D approach as applied in QE. The projector augmented-wave pseudopotentials[86,87] were used. For the relaxation of the crystal, a cutoff of 1000 eV for the plane-wave basis set was used. The relaxation was stopped if the total energy change between two steps for electronic and ionic motions were smaller than $10^{-5}$ and $0.5 \times 10^{-2}$ eV, respectively. Phonon calculations were performed at the Γ-point using a supercell of at least 10 Å length in each direction. Similarly to the relaxation step, a cutoff of 1000 eV for the plane-wave basis set and a convergence threshold of $10^{-8}$ eV were used in the total energy calculation. Next, ZPVE and thermal effects were calculated on a mesh of $8 \times 8 \times 8$ using the dynamical matrix built from the force constants of the displaced atoms in the supercell.

## Data availability

The data that support the findings of this study are included within the Article and Supplementary Information. In particular, the .zip file contains coordinates and energies of all computed data points, parameters of the fits, and the crystallographic information files for a set of top-ranked polymorphs.

## Code availability

The codes used for electronic structure calculations, fitting, CSPs, and pDFT+D calculations: SAPT, ORCA, autoPES (part of the SAPT package), MOLPAK, UPACK, Quantum Espresso, and VASP are available on the web and the links are provided in references of the main paper and the Supplementary Information. A patch to UPACK is available on the SAPT web site. A FORTRAN program computing the fitted potentials is included in the Supplementary_Data_1.zip file.

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

## Acknowledgements

This work was supported by the U.S. Army Research Laboratory and Army Research Office (Grant No. W911NF-19-1-0117) and the NSF (Grant No. CHE-1900551). We thank Rafał Podeszwa for comments on the manuscript.

## Author contributions

R.N. and K.S. designed the method. R.N. coded it and performed numerical calculations. Both authors analyzed the results, wrote the manuscript, and revised it.

## Competing interests

The authors declare no competing interests.
