## [Peer Review File · Nature Communications]

REVIEWER COMMENTS

Reviewer #1 (Remarks to the Author):

The manuscript reports a method for crystal structure prediction of small, rigid molecules using a force field that has been fitted to high level quantum mechanical data. The approach seems to be computationally efficient and offers a dramatic speedup compared to full ab initio approaches (although such methods are rarely used on their own for organic molecular crystal structure prediction). The authors describe the approach, its application to a set of 15 molecules with known crystal structures and an analysis of computational cost.

I like this study. It is a good contribution in the field of crystal structure prediction. However, I think that the authors have exaggerated the impact that this work will have, and the improvement that it offers over existing methods.

1. One problem is that a similar approach exists and has been published: reference 23 (Tailor-made force fields for crystal structure prediction) describes an approach to fit a force field to data derived solely from electronic structure calculations, whose validation is presented in ref 22. I do not know why the current manuscript describes these as "empirical FFs tuned for the system considered to DFT+D calculations for dimers of the molecules involved." As far as the publications describe, these are force fields that are fitted solely to ab initio data. Note also that the methods described in ref 23 are applied to flexible molecules, unlike the rigid molecule validation that is presented in the current manuscript. Note that refs 22 and 23 are from 2008. This is not new methodology. The details are different, such as the use of SAFT(DFT) for generating reference data. Also, it is welcome that the methods described here use open source software, making it more widely available, but this on its own does not lead me to think that the work is of a level required for Nature Communications. Also, given the results presented in ref 22, the statement "Such high level of predictivity is unprecedented for a complete first-principles CSP protocol" is not appropriate.

2. Expanding on the first point, the validation here involves molecules that are too simple to make the claim that the method "will be a transformative change of the whole field of CSPs" (from the conclusions section). Their introduction discusses the blind tests of crystal structure prediction, which have moved on from such rigid molecules. The method here assumes rigid molecules: their geometries are held fixed at the optimized geometries of the isolated monomers. While the discussion states that the method can be extended to flexible molecules, this is not demonstrated. I am less certain that the extension to flexible molecules will be as straightforward as the authors claim. The current 7th blind test of crystal structure prediction (which is ongoing) does not involve any rigid molecules and the 6th blind test, which the authors discuss in their introduction, involves

mostly flexible molecules. A method claiming to be transformative in this area must be demonstrated on molecules that are of a similar complexity to those in the blind tests, or be more qualified in how the method is presented. Methods for rigid molecules can be useful in some application areas of CSP, where rigid molecules are typically used, but then the introduction and discussion needs to make these limitations clearer.

3. The results, before re-optimization of structures using pDFT-D, are not very different from what can be achieved with good quality empirical force fields. Comparing the rankings of the observed crystal structures ("The CSP(aiFF) protocol ranked the experimental polymorph as number 1 in 5 cases, as number 2-6 in 7 cases, and as numbers 9, 9, and 16.") to validation studies with empirical force fields that involve high quality electrostatic models (see, for example, *Crystal Growth & Design* 2005, 5, 3, 1023–1033, <https://doi.org/10.1021/cg049651n>) show similar distributions of rankings. The study presented in *Crystal Growth & Design* 2005, 5, 3, 1023–1033 looked at a larger set of molecules of similar size to those studied here and found that approximately a third of observed crystal structures were ranked as #1 in CSP and two thirds of observed structures in the top 5 ranked structures. This is similar to what has been presented here. There might be a modest improvement, but I am not sure that this would be significant based on 15 molecules. The important advantage of the current approach is that it can be applied equally well to molecules with atom types or functional groups that are not well represented in empirically derived force fields (it is unfortunate that the molecules studied here do not contain any such examples and would all be modelled quite well with empirical force fields). However, for an empirical force field, this can be addressed by a one-off extension of the parametrization.

4. The cost analysis (centered on Fig 3) seems unfair when compared to existing methods. The alternative strategies are set up to be more expensive. For example, "Strategy 2 differs from Strategy 1 by the use of an empirical FF in the PACK+MIN stage and by performing pDFT+D calculations for 100 polymorphs with reoptimization of geometries, a strategy similar to that used in Ref. 41." The complexity of the systems studied in ref 41 is much higher than those studied here, so it is natural that more crystal structures must be considered in the final stages of optimization. This has to do with the complexity of the system rather than the method: molecule XXII from the blind tests is common to both studies (the current manuscript and ref 41) and did not require so many structures in ref 41. However, all other systems in ref 41 feature highly flexible molecules or multicomponent crystals, which are not studied in the present work. It is possible (and I think likely) that application of the current method to the other structures studied in ref 41 would lead to larger numbers of crystal structures within the energy region of uncertainty in the model, so more than 20 re-optimizations would be required. Also, "Strategy 3 performs pDFT+D calculations with geometry optimization for all 25,500 polymorphs produced by PACK+MIN." People don't do this for organic molecular CSP, as everyone knows that it is too expensive. The authors point to references 44 and 45, where pDFT is applied to all structures in CSP, as justification for including this comparison. However, ref 45 studied very simple crystal structures: elemental Li, Na, Mg and Al. These are so simple that it is affordable to optimize all structures using high quality energy calculations (they only studied 1000 crystal structures of each system and the crystal structures are much smaller, so the

associated cost is much lower than for molecular crystals). The comparison is not relevant to molecular organic CSP.

Overall, the study presents a useful method for CSP of organic molecules. It is clearly a useful method that performs well on some fairly simple test systems. However, it is not the level of advance that the authors claim and the comparison to existing methods is not fair in many places. I suggest that the work is more appropriate for a more specialized physical chemistry or materials chemistry journal and that the authors are more careful with their comparison to existing approaches. I would also like to see validation on systems that highlight the strength of the method more strongly: molecules with atoms that are not well-represented by empirical force fields and molecules with intramolecular flexibility, where force fields also have important limitations in their accuracy.

Reviewer #2 (Remarks to the Author):

This paper represents a culmination of many research pieces the Szalewicz group has been working on for years toward the goal of crystal structure prediction (CSP). The key idea here is that they initially parameterize an ab initio force field based on symmetry adapted perturbation theory using a density functional theory description of the monomers: SAPT(DFT). The authors have published a number of papers along these lines, but this paper stands out in how they have automated the procedures, have made them fast enough to perform routinely, and have therefore been able to predict the crystal structures for 15 small molecules with good success. In other words, this is getting much closer to a "black box" procedure than in their earlier work. Organic materials have many important applications, and the ability to predict their structures routinely in the future will have a profound effect on our ability to design them rationally.

This work presents a nice demonstration of how these types of methods have advanced in recent years, but there remain non-trivial limitations here. Most notably, the molecules are relatively easy, in the sense that they are nearly or entirely rigid. Conformational polymorphs of flexible molecules are far harder to predict correctly, and many of the applications where CSP is used exhibit considerable flexibility. This and other limitations are acknowledged in "Neglected effects" section, which I appreciate. I am perhaps not quite as optimistic as they are about how directly this present success will translate to those molecules with the simplifications it will require, but that's the subject for a future paper (and they may well prove me wrong!). Regardless, these limitations do not detract from the current scope and accomplishments of the paper.

Overall, the paper is nicely done and the results are impressive. It deserves publication after the minor revisions described below.

- One strength of the work that is perhaps not emphasized strongly enough is how the use of the SAPT(DFT) potential throughout the early CSP stages is a key advantage compared to traditional CSP approaches. Traditional CSP work flows are hierarchical, with inexpensive, low-accuracy methods used in the early stages, followed by subsequent stages of refinement with more accurate and expensive models on only the most promising candidates from each previous stage. But there have been plenty of cases where correct structures were discarded erroneously in the intermediate stages due to the limitations of the lower-accuracy models. The high-quality SAPT-based potentials are promising for avoiding those pitfalls.

- I am surprised that their potential works as well as it does, given the use of only atomic charges instead of higher-order multipoles. A lot of earlier CSP work by Stone, Price, and others found multipolar potentials to be important (e.g. Price's 2008 review, DOI: 10.1080/01442350802102387). Do the authors have thoughts on this?

- Similarly, the authors do acknowledge the neglect of many-body effects here, but I am again surprised that this is not a bigger limitation in these systems. A lot of classic work from Price (e.g. DOI: 10.1021/ct700270d, 10.1063/1.2937446) highlights the importance of induction energies in CSP. More recently, DOI: 10.1039/c9sc05689k found that despite accounting for only ~5% of the lattice energy in oxalyl dihydrazide, many-body effects constituted a much larger fraction of the relative lattice energies. Other many-body effects like dispersion/exchange might become important in cases where packing densities differ substantially between forms (e.g. high-pressure polymorphs). The results in the current paper speak for themselves, but it could be worthwhile mentioning counter-examples where many-body effects are important in the "Neglected effects" section.

- Separately, I think they should make the fact that they are using a 2-body, non-polarizable fitted potential clear much earlier in the paper.

- pg 3: In the "Importance of using extended form" section, it would be helpful to explain qualitatively/physically what the better form includes. Instead, I found myself jumping between this section and Methods to try to understand what they were saying. The section title here is also overly vague. Maybe something more like "Importance of the potential functional form" would be clearer?

- pg 4: Cost comparisons: It would be helpful to emphasize that Strategies 2 & 3 are hypothetical approaches one might consider. At first, I was wondering if I had missed an earlier part of the paper

where those were defined, or if they corresponded to the strategies in the "Alternative CSP(aiFF) protocol" section, and to use conditional tenses/phrasing ("cost would be an order of magnitude more expensive" or similar).

- pg 5: "These effects can be routinely computed using pDFT+D, although such calculations are several times more expensive than

pDFT+D calculations with static geometries." This sentence is unclear to me. Are they comparing periodic DFT single-point

energies versus geometry optimizations? Or are they talking about rigid-molecule periodic DFT optimizations?

- For the sake of reproducibility, testing of other methods on these systems, etc, it would be good to provide energies and crystal structures for each of the CSP searches. The information provided does not presently enable one to validate the results presented. Perhaps they might consider archiving the training data and fitted potentials somehow as well? (the latter might be more suitable for an online repository)

Finally, a few minor aspects in the work are over-sold, in my opinion:

- pg 2: "The variant of SAPT used by us (see Methods) is comparably accurate to the coupled cluster method with single, double, and noniterative triple excitations, CCSD(T)" I think highly of SAPT(DFT), but this phrasing to me suggests a parity between SAPT(DFT) and CCSD(T) that I do not think is supported by available benchmarks. Moreover, here the authors are only using an aug-cc-pVDZ basis set, which is far from converged. That basis set is helped by the addition of mid-bond basis functions. Still, in Ref 35, for example, the SAPT(DFT) errors with the larger aug-cc-pVTZ basis in Fig 3 are only slightly better than MP2 or B3LYP-D3. Again, I don't intend to be critical of SAPT(DFT), but it should not be over-sold.

- In addition, the authors should take care not to conflate the accuracy of SAPT(DFT) for a molecular dimer with that of the description of the many-body crystal. Even if SAPT(DFT) managed to mimic CCSD(T) perfectly for the dimer interactions, it is not achieving that accuracy for the full crystal due to the neglect of many-body effects (of course, CCSD(T) isn't realizing that accuracy either due to computational cost!). The authors should be more precise in this context.

- pg 4: The paper mentions a fit error of ~ 1 kJ/mol. I believe this is referring to errors in the pairwise intermolecular interactions. Similar to the previous point, the crystal involves many such pairwise interactions, and the errors will frequently compound in the lattice energy. I don't think this is catastrophic, since there will likely be considerable error cancellation between different polymorphs, but it claimed error should be clarified.

Reviewer #3 (Remarks to the Author):

The manuscript 'Reliable crystal structure predictions from first principles' introduces a new molecular crystal structure prediction protocol and benchmarks this approach for several molecular crystals involving mainly rigid molecules. This approach utilizes an ab initio-based force field, which is fitted to symmetry-adapted perturbation theory (SAPT) calculations. The authors show that this new protocol is able to predict the experimentally observed polymorph within the top 20 structures for all systems investigated and the authors could further improve the ranking by subsequent DFT calculations. This manuscript will be an important contribution to the field of crystal structure prediction.

Therefore, I recommend the publication of this manuscript after the following comments have been addressed:

1.) The authors use 8 systems from previous blind tests and 7 other molecular crystals to validate their procedure. Until now, the published blind tests included 26 different systems. Therefore, the authors should comment in the main text on why only these 8 particular systems were selected. The authors seem to focus only on single-component crystals involving relatively small and rigid molecules, which is of course reasonable for an initial validation of a new procedure. However, the performance for co-crystals, salts, hydrates, and crystals involving larger and flexible molecules cannot really be assessed yet. Therefore, the authors should also explicitly mention in the main text and the conclusion which types of molecular crystals were benchmarked.

2.) It might be beneficial for several readers to briefly also mention how much more expensive the newly developed force field is compared to the typical force fields used in CSP.

3.) Please add some computational details to the description of the ZPVE and thermal effects calculations. Were they performed directly in VASP or with an external tool like phonopy? In case of finite differences, which supercells and q-grids were used?

Report of Reviewer 1

The manuscript reports a method for crystal structure prediction of small, rigid molecules using a force field that has been fitted to high level quantum mechanical data. The approach seems to be computationally efficient and offers a dramatic speedup compared to full ab initio approaches (although such methods are rarely used on their own for organic molecular crystal structure prediction). The authors describe the approach, its application to a set of 15 molecules with known crystal structures and an analysis of computational cost.

I like this study. It is a good contribution in the field of crystal structure prediction. However, I think that the authors have exaggerated the impact that this work will have, and the improvement that it offers over existing methods.

We thank the reviewer for an accurate characterisation of our work and for the generally positive evaluation of it. We will answer the criticism concerning our statements on impact of our work and on the improvement it offers over the existing approaches at the end of our response since this criticism appears to be related to other comments of the reviewer.

1. One problem is that a similar approach exists and has been published: reference 23 (Tailor-made force fields for crystal structure prediction) describes an approach to fit a force field to data derived solely from electronic structure calculations, whose validation is presented in ref 22. I do not know why the current manuscript describes these as “empirical FFs tuned for the system considered to DFT+D calculations for dimers of the molecules involved.” As far as the publications describe, these are force fields that are fitted solely to ab initio data. Note also that the methods described in ref 23 are applied to flexible molecules, unlike the rigid molecule validation that is presented in the current manuscript. Note that refs 22 and 23 are from 2008. This is not new methodology.

We believe the approach of Neumann *et al.* (Refs. 22 and 23) is significantly different from our approach, in fact, in our opinion it is not similar at all. We have extended the description of this method on page 2 to emphasize the differences. First, as we say in the manuscript, while the former approach is the most successful CSP method on the market, it cannot be considered as a fully first-principles method, i.e., we cannot agree with the reviewer that the tailor-made FFs of Neumann *et al.* are fit “to data derived solely from electronic structure calculations”, while our approach is of this type. The starting point for a tailor-made FF is the Dreiding empirical FF (see Ref. 23, p. 9812). The parameters of the Dreiding FF are, as stated several times in Ref. 23, “refined” using *ab initio* lattice energies and their derivatives (the use of the word “dimers” in our text was a mistake which has been fixed). In contrast, aiFF uses no experimental information and fits all parameters from scratch. FFs that are obtained by modifications of empirical FFs are often called “tuned empirical FFs” and this was the phrase used by us. We would like to emphasize that the fact that the tailor-made FFs of Neumann *et al.* are not obtained fully from first principles does not constitute any criticism of this method, but does point out to a significant difference between the two approaches. The second difference is that the *ab initio* data that are used to tune tailor-made FFs are obtained from pDFT+D calculations on crystals, while CSP(aiFF) uses SAPT calculations for dimers. While

each approach has its advantages, the differences between them are significant. One may also add that to get the initial crystal structures for the pDFT+D calculations in the method of Neumann *et al.* an empirical FF has to be used. Furthermore, the pDFT+D method used by Neumann *et al.* does have an empirical flavor and in fact the title of the Neumann-Perrin paper in *J. Phys. Chem. B* **109**, 15531 (2005) is: “Energy Ranking of Molecular Crystals Using Density Functional Theory Calculations and an Empirical van der Waals Correction”. Finally, some details of the Neumann *et al.* approach have not been published, no source codes are available, and only commercial software realizing this approach exists. Although as the reviewer stated, this methodology was published in 2008, to our knowledge, there was no effort by other researchers to create an open-source version.

The fact that the method of Neumann *et al.* applies to flexible monomers is indeed an advantage over the aiFF approach presented in our manuscript. We discussed this restrictions and ways to extend the CSP(aiFF) method to flexible monomers in the section “Neglected effects” on page 5 (this text has now been expanded). We are currently working on such an extension, but we believe that crystals with rigid molecules are an important enough and large class of crystals to make our method relevant for the field. In the past, CSPs for such molecules were a laborious and daunting task, while we show that such CSPs can be easily performed with CSP(aiFF).

The details are different, such as the use of SAPT(DFT) for generating reference data. Also, it is welcome that the methods described here use open source software, making it more widely available, but this on its own does not lead to me think that the work is of a level required for Nature Communications. Also, given the results presented in ref 22, the statement “Such high level of predictivity is unprecedented for a complete first-principles CSP protocol” is not appropriate.

We hope that our arguments given above will convince the reviewer that CSP(aiFF) is a method fundamentally different from the method of Neumann *et al.* and only the former method is a fully first-principles one. If this is the case, the sentence quoted is actually accurate. Indeed, if the differences were only due to our use of SAPT and providing open-source codes, we would not have sent our manuscript to Nature Communications. However, the differences are substantial as described above. We believe the importance of our work stems from combining recent developments in *ab initio* methodologies and their numerical implementations [points (a)-(d) described on page 2 of our manuscript] into a reliable CSP protocol.

2. Expanding on the first point, the validation here involves molecules that are too simple to make the claim that the method “will be a transformative change of the whole field of CSPs” (from the conclusions section).

As stated above, the class of rigid molecules is actually important on its own. We could have included larger molecules but since there are no reasons to doubt that the method will work for any size molecule that one can afford computationally (in fact, our group is now developing an aiFF for a 100-atom monomer), there was no point to do so. In our opinion, the molecules included provide sufficient evidence that one can expect “transformative change”, i.e., the field of CSPs will transform from the use of empirical FFs to first-principle ones (we have clarified our use of the word “transformative” in the

Conclusions). Although in the work presented in the manuscript we have not applied aiFF to crystals with flexible monomers, our results for the rigid ones indicate that an extended CSP(aiFF) method should work well for flexible monomers. The reason is that the flexible-monomer empirical FFs consist of uncoupled intermonomer and intramonomer parts. The former are, for a given geometry of a monomer, exactly the same as our rigid-monomer aiFFs. Thus, replacement of the intermonomer parts of empirical FFs by aiFFs should improve accuracy of such FFs.

Their introduction discusses the blind tests of crystal structure prediction, which have moved on from such rigid molecules. The method here assumes rigid molecules: their geometries are held fixed at the optimized geometries of the isolated monomers. While the discussion states that the method can be extended to flexible molecules, this is not demonstrated. I am less certain that the extension to flexible molecules will be as straightforward as the authors claim.

We are currently working on such an extension and the results are encouraging. As stated in the manuscript on page 5, we assume that the intermolecular and intramolecular degrees of freedom are uncoupled, as is the case for all empirical FFs and also for the tailor-made FFs of Neumann *et al.* Since the analytic form of our aiFF depends only on atom-atom distances, the interaction energy can be computed for any deformations of monomers. We call an aiFF developed for rigid monomers and used for different monomers geometries a “flexibilized” aiFF. This intermonomer FF is used together with the intramonomer term of an empirical FF regularized to have the minimum at the *ab initio* equilibrium geometry. We apply UPACK to perform flexible-monomer CSPs with such a hybrid FF. Since the quality of predictions of flexibilized aiFFs decreases with increased deformation of monomers, this quality is monitored by performing SAPT(DFT) calculations for nearest neighbour dimers in polymorphs predicted by UPACK. If the difference between interaction energies from aiFF and from SAPT(DFT) becomes larger than some assumed threshold, a new aiFF is developed using deformed monomers found by UPACK. This is less expensive than the development of the original aiFF since the previously computed grid points can be reused. A condensed version of these explanations was added on page 5.

The current 7th blind test of crystal structure prediction (which is ongoing) does not involve any rigid molecules and the 6th blind test, which the authors discuss in their introduction, involves mostly flexible molecules. A method claiming to be transformative in this area must be demonstrated on molecules that are of a similar complexity to those in the blind tests, or be more qualified in how the method is presented. Methods for rigid molecules can be useful in some application areas of CSP, where rigid molecules are typically used, but then the introduction and discussion needs to make these limitations clearer.

We are actually taking part in the 7th blind test and we have submitted predictions for all 7 targets. Furthermore, as described above, we do believe that our results for rigid monomers are indicative of future applications of aiFFs to flexible monomers. We have made several changes in the manuscript to make the current limitations of the CSP(aiFF) method clearer. In particular, we added an explicit statement on the restriction to rigid monomers in the abstract, extended the text on page 5 on future applications to flexible

monomers, and repeated this information in the conclusions.

3. *The results, before re-optimization of structures using pDFT-D, are not very different from what can be achieved with good quality empirical force fields. Comparing the rankings of the observed crystal structures (“The CSP(aiFF) protocol ranked the experimental polymorph as number 1 in 5 cases, as number 2-6 in 7 cases, and as numbers 9, 9, and 16.”) to validation studies with empirical force fields that involve high quality electrostatic models (see, for example, Crystal Growth & Design 2005, 5, 3, 1023–1033, <https://doi.org/10.1021/cg049651n>) show similar distributions of rankings. The study presented in Crystal Growth & Design 2005, 5, 3, 1023–1033 looked at a larger set of molecules of similar size to those studied here and found that approximately a third of observed crystal structures were ranked as #1 in CSP and two thirds of observed structures in the top 5 ranked structures. This is similar to what has been presented here. There might be a modest improvement, but I am not sure that this would be significant based on 15 molecules. The important advantage of the current approach is that it can be applied equally well to molecules with atom types or functional groups that are not well represented in empirically derived force fields (it is unfortunate that the molecules studied here do not contain any such examples and would all be modelled quite well with empirical force fields). However, for an empirical force field, this can be addressed by a one-off extension of the parametrization.*

We cannot agree with Reviewer’s 1 statement that the quality of CSPs with empirical force fields “is similar to what has been presented here”. The data from the paper by Day *et al.* in CG&D 2005, quoted by the reviewer, should not be compared with our rankings, in fact, they are not rankings. Our rankings are obtained in blind-test fashion, i.e., using no experimental information. Then the purely geometrical RMSD₂₀ criterion is used to find which predicted polymorphs are close to the experimental ones. Day *et al.* did not perform the second of these steps. Instead, they compare their results to “the experimentally observed structures after energy minimization with the W99 + multipoles model potential”. Day *et al.* did not report what is the rank of the experimental polymorphs among theoretically predicted polymorphs. The quantity N_{lower} given in their Table 1 is just the number of theoretical polymorphs with the energies below the experimental polymorph. These numbers are not necessarily indicative of a priori rankings and N_{lower} should not be compared with our rankings.

The comment of Reviewer 1 prompted us to check how good are the predictions of the empirical FF (W99+point-charges model) used by Day *et al.* compared to aiFFs. We obtained the point charges using the CHELPG method [Breneman and Wiberg *J. Com. Chem.* **11**, 361 (1990)]. The results listed in Table 1 below demonstrate that the performance of aiFFs is significantly better, in fact, qualitatively better. For example, aiFF has 94% of cases ranked at positions 10 or better, while W99+charges only 33%. This is indeed a qualitative difference for technological applications. These results are described in a new section “Performance of an empirical FF” on page 4 and in SI.

The work of Day *et al.* has shown that an electrostatic model using atomic multipoles performs better than charges-only model, but the differences between the two models are much smaller than the difference between W99+charges and aiFF, as shown in Table 2 below.

Table 1: Comparison of ranking of the 18 polymorphs by aiFF and by W99+charges FF. The numbers give the percent of the polymorphs at a given range of ranks, with cumulative values in parentheses. MS denotes missing parameters in W99 FF. The column >100 counts CSPs that did not include the experimental polymorph within 3112, 568, 1272, and 2463 lowest lattice energy polymorphs for monomers I, II, benzene, and deferiprone, respectively.

	1	2-10	11-20	21-100	>100	MS
aiFF	28	67 (94)	6 (100)			
W99+charges	6	28 (33)	11 (44)	22 (67)	22 (89)	11 (100)

Table 2: Comparison of charges-only and multipole moments models from Table 1 in Day *et al.* The percentages are computed using all 65 rows in this table.

N_{lower}	1	≤ 5	≤ 10	≤ 20
charges	29	53	69	81
multipoles	34	65	82	89

Reviewer 1 wrote that it is unfortunate that our set of molecules does not include atom types or functional groups that are not well represented in empirically derived force fields. Our results with the W99+charges show that actually there are several such examples. For two molecules the CSPs could not be done since the atom types needed are not available in W99+charges. CSPs based on W99+charges missed 4 out of 18 polymorphs completely (these polymorphs were not present on the list of polymorphs that were examined). Clearly, some functional groups in these molecules are poorly represented by the W99+charges FF. One may add that molecules like DNBT and TNB are not well parametrized in most empirical force fields. The W99+charges FF actually performs for these crystals better than other empirical FFs since the training set of W99 included a large number of nitrogen-containing molecules. Still, the RMSD₂₀ for DNBT is 0.81 Å, slightly above the CCDC threshold of 0.8 Å.

4. The cost analysis (centered on Fig 3) seems unfair when compared to existing methods. The alternative strategies are set up to be more expensive. For example, “Strategy 2 differs from Strategy 1 by the use of an empirical FF in the PACK+MIN stage and by performing pDFT+D calculations for 100 polymorphs with reoptimization of geometries, a strategy similar to that used in Ref. 41.” The complexity of the systems studied in ref 41 is much higher than those studied here, so it is natural that more crystal structures must be considered in the final stages of optimization. This has to do with the complexity of the system rather than the method: molecule XXII from the blind tests is common to both studies (the current manuscript and ref 41) and did not require so many structures in ref 41.

First, Strategies 2 and 3 were not intended to criticize existing methods. We mentioned

such methods just to show that similar approaches have been used. We have now removed these mentions and stated that these strategies are hypothetical. With the new results obtained using the W99+charges FF, we were able to evaluate these strategies in more precise terms. In particular, we can now state that the success rate of Strategy 2 is 67%, see Table 1 here.

We also have now realized that the comparison with the work of Hoja *et al.*, Ref. 41, was inappropriate, but for reasons different than given by the reviewer: the 100 polymorphs were obtained in that work not from an empirical FF, but using the tailor-made FFs of Neumann *et al.*, which are, of course, much more predictive than any empirical FF. This is the reason why, as the reviewer rightly points out to, the 100 polymorphs were sufficient for crystals with flexible monomers.

BTW, we are not sure why the reviewer implies that Hoja *et al.* used less than 100 structures for system XXII. What they say is: “*As the foundation for the presented stability ranking approach, we use the top 100 molecular crystal structures (for every system of the latest blind test) from the above mentioned sampling approach of Neumann and co-workers using GRACE.*”

However, all other systems in ref 41 feature highly flexible molecules or multicomponent crystals, which are not studied in the present work. It is possible (and I think likely) that application of the current method to the other structures studied in ref 41 would lead to larger numbers of crystal structures within the energy region of uncertainty in the model, so more than 20 re-optimizations would be required.

It is possible that flexible monomers will require more than 20 structures. However, any CSPs with flexible monomers have to be significantly more expensive than with rigid monomers, this is simply due to the increased dimensionality which affects all CSP methods.

Also, “Strategy 3 performs pDFT+D calculations with geometry optimization for all 25,500 polymorphs produced by PACK+MIN.” People don’t do this for organic molecular CSP, as everyone knows that it is too expensive. The authors point to references 44 and 45, where pDFT is applied to all structures in CSP, as justification for including this comparison. However, ref 45 studied very simple crystal structures: elemental Li, Na, Mg and Al. These are so simple that it is affordable to optimize all structures using high quality energy calculations (they only studied 1000 crystal structures of each system and the crystal structures are much smaller, so the associated cost is much lower than for molecular crystals). The comparison is not relevant to molecular organic CSP.

The goal of considering Strategy 3 was to see how expensive would be CSPs based on empirical FFs which would be nearly 100% successful. We have revised the text to make it clear why we consider these strategies. Citing Refs. 44 and 45, we did say that this work was for smaller monomers than considered by us. Still, the idea is the same. Furthermore, an approach somewhat similar to Strategy 3 was used in the 6th blind test by Maron *et al.* For target XXII, they performed 200,000 pDFT+D calculations, although with a very approximate DFT functional. In any case, we have deleted these references and now discuss Strategy 3 as a hypothetical one.

Overall, the study presents a useful method for CSP of organic molecules. It is clearly a

useful method that performs well on some fairly simple test systems. However, it is not the level of advance that the authors claim and the comparison to existing methods is not fair in many places. I suggest that the work is more appropriate for a more specialized physical chemistry or materials chemistry journal and that the authors are more careful with their comparison to existing approaches. I would also like to see validation on systems that highlight the strength of the method more strongly: molecules with atoms that are not well-represented by empirical force fields and molecules with intramolecular flexibility, where force fields also have important limitations in their accuracy.

We will first respond to the statements of Reviewer 1 from the opening paragraphs: “However, I think that the authors have exaggerated the [...] the improvement that [their method] offers over existing methods” and the statement made above “it is not the level of advance that the authors claim”. The *level of advance* of our work is determined by our combination of a reliable and efficient *ab initio* method with a method for automatic fitting of *ab initio* interaction energies and then the use of the resulting aiFFs (of a form more elaborate than that of most empirical FFs) in CSPs, followed by reranking with pDFT+D. We believe this is a true advance and no similar protocol has been available, therefore, our work does constitute a significant *improvement over existing methods* in terms of theory development. In particular, we hope we have convinced the reviewer that our method is not similar to that of Neumann *et al.* An important difference between the two approaches is that the latter is partly based on empirical input, while our is a completely first-principles method. We believe that moving from empiricism to fully first-principles treatments is an important advancement in any field of science.

The next criticism is that *the comparison to existing methods is not fair in many places*. One of those comparisons was the relation between the Neumann *et al.* and our method, already discussed. Another comparison concerned the improvements of predictability of the pure aiFF approach to that of empirical FFs, which the reviewer considers to be only “modest” based on comparisons to Day *et al.* We first point out that the quantities that the reviewer extracted from the Day *et al.* paper are not rankings of a blind search. We then performed CSPs on our set of molecules applying the W99+charges FF used in Day *et al.* The results show that the improvement due to the use of aiFF is not modest, but quite dramatic, really moving CSPs to a new level of quality of predictions. This comparison also shows that we have not *exaggerated the improvement CSP(aiFF) offer over existing methods*. Finally, the reviewer believes our costs comparisons are unfair. This is a misunderstanding, we had no intention to compare costs to existing methods. It is now clearly stated in the manuscript that the methods considered in costs comparisons are hypothetical.

The criticism that we do not provide examples how our method works on “molecules with atoms that are not well-represented by empirical force fields” has been answered by our application of the W99+charges empirical FF to our set of molecules: W99+charges fails really badly on several of them.

We cannot provide examples of how CSP(aiFF) works for “molecules with intramolecular flexibility” since the current version of our CSP protocol works only for rigid monomers (aiFFs have been applied in molecular dynamics simulations with flexible monomers, for example in the 6th blind test, but this protocol is different from what we propose here). The extension to flexible monomers is under development in our group. However, as we

argued above, the class of rigid monomers is, in our opinion, broad enough for our method to be of significant importance. Moreover, as we also discussed above, there are good reasons to believe that the use of aiFFs in CSPs with flexible monomers will also result in improved predictions.

Finally, let us address the criticism from the opening paragraphs that we “have exaggerated the impact that this work will have”. We do believe the impact will be significant for several reasons. First, we have developed the first fully first-principles CSP method which can be used in an automated fashion. Second, CSP(aiFF) performs dramatically better than similar approaches based on empirical FFs. Moving from empiricism to first principle methods is an important advancement in any field of science. Third, although the method works now only for rigid monomers, it is evident that its extension should improve flexible-monomer CSPs. While the reviewer criticized our statement that “the proposed CSP protocol should result in a transformative change of the field of CSPs”, we believe the impact of our work will be that researchers working on CSPs will start using aiFFs instead of empirical FFs. Such a transformation has already happened in the field of van der Waals clusters and in molecular dynamics simulation of fluids of small molecules. This transformation will be helped by the fact that the CSP(aiFF) protocol is an easy to use (since it is highly automated) approach freely available to all researchers. Until now, there has not been any method on the market with all these attributes.

Report of Reviewer 2

This paper represents a culmination of many research pieces the Szalewicz group has been working on for years toward the goal of crystal structure prediction (CSP). The key idea here is that they initially parameterize an ab initio force field based on symmetry adapted perturbation theory using a density functional theory description of the monomers: SAPT(DFT). The authors have published a number of papers along these lines, but this paper stands out in how they have automated the procedures, have made them fast enough to perform routinely, and have therefore been able to predict the crystal structures for 15 small molecules with good success. In other words, this is getting much closer to a “black box” procedure than in their earlier work. Organic materials have many important applications, and the ability to predict their structures routinely in the future will have a profound effect on our ability to design them rationally.

This work presents a nice demonstration of how these types of methods have advanced in recent years, but there remain non-trivial limitations here. Most notably, the molecules are relatively easy, in the sense that they are nearly or entirely rigid. Conformational polymorphs of flexible molecules are far harder to predict correctly, and many of the applications where CSP is used exhibit considerable flexibility. This and other limitations are acknowledged in “Neglected effects” section, which I appreciate. I am perhaps not quite as optimistic as they are about how directly this present success will translate to those molecules with the simplifications it will require, but that’s the subject for a future paper (and they may well prove me wrong!). Regardless, these limitations do not detract from the current scope and accomplishments of the paper.

Overall, the paper is nicely done and the results are impressive. It deserves publication after the minor revisions described below.

We thank the reviewer for such a positive opinion about our work.

- *One strength of the work that is perhaps not emphasized strongly enough is how the use of the SAPT(DFT) potential throughout the early CSP stages is a key advantage compared to traditional CSP approaches. Traditional CSP work flows are hierarchical, with inexpensive, low-accuracy methods used in the early stages, followed by subsequent stages of refinement with more accurate and expensive models on only the most promising candidates from each previous stage. But there have been plenty of cases where correct structures were discarded erroneously in the intermediate stages due to the limitations of the lower-accuracy models. The high-quality SAPT-based potentials are promising for avoiding those pitfalls.*

We fully agree fully with the reviewer and added an appropriate statement in Conclusions.

- *I am surprised that their potential works as well as it does, given the use of only atomic charges instead of higher-order multipoles. A lot of earlier CSP work by Stone, Price, and others found multipolar potentials to be important (e.g. Price’s 2008 review, DOI: 10.1080/01442350802102387). Do the authors have thoughts on this?*

Actually, our electrostatic energies are quite accurate despite using only charges. We do use multipoles in the asymptotic range. Our procedurs represents the total monomer density $\rho(\mathbf{r})$ as the sum of atomic densities $\rho_a(\mathbf{r})$. The latter densities are used to compute a set of multipole moments on each atom, which are subsequently used to compute electrostatic energies at asymptotic separations only. Such electrostatic energies agree very well in that region with those obtained from *ab initio* multipole moments located at the center of mass of each monomer. We then fit this set of electrostatic energies using only Coulomb interactions of optimized partial charges. Such atomic charges reproduces asymptotic electrostatic energies with an error of about 5%. These charges are kept fixed at close-range fitting, but the damping functions in Eq. (1) are optimized. To show how accurate are the electrostatic energies, we have added Fig. S5 in SI showing the radial dependence of the fit components and of the corresponding SAPT(DFT) ones. One can see in particular that the *ab initio* electrostatic energies are reproduced very well for R ’s larger than the van der Waals minimum distance R_{vdW} despite using only damped charge-charge interactions, i.e., omitting higher multipolar terms. While we agree that the use of the latter terms in empirical FFs improves the predictions compared to the use of charges only, our results show that higher-rank multipoles are not needed if the electrostatic function includes damping and is fitted to *ab initio* electrostatic energies. The worsening of the agreement with *ab initio* values seen for $R < R_{\text{vdW}}$ is inevitable and is due to the charge-overlap effects that are not proportional to inverse powers of R (the so-called “spherical” terms in asymptotic expansions). These effects are accounted for in the overall fit by the first term of Eq. (1) in Methods. This is why the total fitted and *ab initio* interaction energies are in excellent agreement for all R . This text was included in Methods at the end of section “Generation of aiFFs”.

- *Similarly, the authors do acknowledge the neglect of many-body effects here, but I am again surprised that this is not a bigger limitation in these systems. A lot of classic work from Price (e.g. DOI: 10.1021/ct700270d, 10.1063/1.2937446) highlights the importance*

of induction energies in CSP. More recently, DOI: 10.1039/c9sc05689k found that despite accounting for only 5% of the lattice energy in oxalyl dihydrazide, many-body effects constituted a much larger fraction of the relative lattice energies. Other many-body effects like dispersion/exchange might become important in cases where packing densities differ substantially between forms (e.g. high-pressure polymorphs). The results in the current paper speak for themselves, but it could be worthwhile mentioning counter-examples where many-body effects are important in the "Neglected effects" section.

We agree that many-body effects in crystal investigated by us are smaller than expected. In the initial stages of this research, our hypothesis was that those are the many-body effects that are responsible for the differences between the predictions from pure aiFFs (Stage 4) and from pDFT+D (Stage 5). However, the performance of the alternative CSP(aiFF) protocol as well as the lack of correlation between rankings and two-body induction effects falsified this hypothesis. Apparently the impact of many-body effects on structure of the investigated crystals is small. Certainly, there will be cases where it has to be more substantial and we are working on including many-body polarization effects in CSPs. We have added citations to the papers mentioned by the reviewer.

- Separately, I think they should make the fact that they are using a 2-body, non-polarizable fitted potential clear much earlier in the paper.

We have added the information that we use only two-body potentials to the abstract (this already implies that we have not used three- and higher-body pairwise nonadditive polarization effects).

- pg 3: In the "Importance of using extended form" section, it would be helpful to explain qualitatively/physically what the better form includes. Instead, I found myself jumping between this section and Methods to try to understand what they were saying. The section title here is also overly vague. Maybe something more like "Importance of the potential functional form" would be clearer?

We have changed the title of the subsection to "Performance of a simplified aiFF form" and extended the first sentence of this section to explain the difference between these two forms.

- pg 4: Cost comparisons: It would be helpful to emphasize that Strategies 2 & 3 are hypothetical approaches one might consider. At first, I was wondering if I had missed an earlier part of the paper where those were defined, or if they corresponded to the strategies in the "Alternative CSP(aiFF) protocol" section, and to use conditional tenses/phrasing ("cost would be an order of magnitude more expensive" or similar).

The suggested changes have been made.

- pg 5: "These effects can be routinely computed using pDFT+D, although such calculations are several times more expensive than pDFT+D calculations with static geometries." This sentence is unclear to me. Are they comparing periodic DFT single-point energies versus geometry optimizations? Or are they talking about rigid-molecule periodic DFT optimizations?

The effects referred to in the quoted sentence are thermal and entropic effects. To compute them, one has to first perform complete optimization of geometry, so that the

forces are zero and the equilibrium structure Hessian can be computed providing the vibrational energies. Such calculation is much more time consuming than a single fixed-geometry calculation of lattice energy. We have modified the quoted sentence to make it more clear. Also, to respond to another reviewer, we have extended the last paragraph of the Methods which describes our calculations of thermal and entropic effects.

- For the sake of reproducibility, testing of other methods on these systems, etc, it would be good to provide energies and crystal structures for each of the CSP searches. The information provided does not presently enable one to validate the results presented. Perhaps they might consider archiving the training data and fitted potentials somehow as well? (the latter might be more suitable for an online repository)

A .zip file with the information required has been included in the resubmission.

Finally, a few minor aspects in the work are over-sold, in my opinion:

- pg 2: "The variant of SAPT used by us (see Methods) is comparably accurate to the coupled cluster method with single, double, and noniterative triple excitations, CCSD(T)" I think highly of SAPT(DFT), but this phrasing to me suggests a parity between SAPT(DFT) and CCSD(T) that I do not think is supported by available benchmarks. Moreover, here the authors are only using an aug-cc-pVDZ basis set, which is far from converged. That basis set is helped by the addition of mid-bond basis functions. Still, in Ref 35, for example, the SAPT(DFT) errors with the larger aug-cc-pVTZ basis in Fig 3 are only slightly better than MP2 or B3LYP-D3. Again, I don't intend to be critical of SAPT(DFT), but it should not be over-sold.

We have changed the phrase "comparably accurate to" to "nearly as accurate as". The previous phrase is actually true, but may depend on how one determines the size of deviations that this phrase encompasses. The most thorough comparison of SAPT(DFT) with CCSD(T) was performed in Ref. 43 on 10 dimers and about 100 configurations total. The median unsigned percentage error computed for all dimers in an augmented triple-zeta basis relative to CCSD(T)/CBS was 2.6%. This should be compared to the same error for CCSD(T) in the same basis set amounting to 1.2%. Thus, the difference in accuracy is 1.4%. In our opinion, this is a "comparable" accuracy on the scale of DFT and DFT+D methods which for the same benchmark set range between 3.4% and 8%. Some modern DFT+D methods are indeed quite accurate, and the best of those included in Ref. 43, B3LYP-D3, could perhaps be considered to be comparable in accuracy to CCSD(T). However, while B3LYP-D3 performs very well on the benchmarks of Ref. 43, this is not necessarily the case for some other benchmarks (see for example Fig. 2 in Ref. 42), so it is usually not included in this category of methods. MP2, with 3.0% error, also performs very well on the set of Ref. 43. However, one cannot state that MP2 is comparably accurate to CCSD(T) since it performs dramatically bad on interactions of aromatic molecules.

The use of the relatively small aug-cc-pVDZ+midbond basis set is an aspect of our work unrelated to SAPT(DFT)/CCSD(T) relative accuracy of the two methods. The effect of basis set truncation was discussed in the SI in the first paragraph of Sec. II. We

have added a pointer to this discussion in the main text.

- In addition, the authors should take care not to conflate the accuracy of SAPT(DFT) for a molecular dimer with that of the description of the many-body crystal. Even if SAPT(DFT) managed to mimic CCSD(T) perfectly for the dimer interactions, it is not achieving that accuracy for the full crystal due to the neglect of many-body effects (of course, CCSD(T) isn't realizing that accuracy either due to computational cost!). The authors should be more precise in this context.

We agree with these statements, but we thought the fact that two-body lattice energies do not constitute the total lattice energies was clearly delineated in our manuscript. To make it even more clear, we added on page 2 that the comparison of SAPT to CCSD(T) applies only to dimer interaction energies and started the section "Neglected effects" from stating that aiFF includes only two-body interactions.

- pg 4: The paper mentions a fit error of 1 kJ/mol. I believe this is referring to errors in the pairwise intermolecular interactions. Similar to the previous point, the crystal involves many such pairwise interactions, and the errors will frequently compound in the lattice energy. I don't think this is catastrophic, since there will likely be considerable error cancellation between different polymorphs, but it claimed error should be clarified.

We agree and again we thought this was obvious. We have added the information that RMSE is computed for dimer configurations. There is, indeed, cumulation of absolute errors, but the relative error of lattice energy is of the same order of magnitude as for the dimer since both quantities add up: $\sum (E_{\text{int},i} + \delta_i) = \sum E_{\text{int},i} + \sum \delta_i$.

Report of Reviewer 3

The manuscript 'Reliable crystal structure predictions from first principles' introduces a new molecular crystal structure prediction protocol and benchmarks this approach for several molecular crystals involving mainly rigid molecules. This approach utilizes an ab initio-based force field, which is fitted to symmetry-adapted perturbation theory (SAPT) calculations. The authors show that this new protocol is able to predict the experimentally observed polymorph within the top 20 structures for all systems investigated and the authors could further improve the ranking by subsequent DFT calculations. This manuscript will be an important contribution to the field of crystal structure prediction.

Therefore, I recommend the publication of this manuscript after the following comments have been addressed:

We thank the reviewer for his or her appreciation of our work.

1.) The authors use 8 systems from previous blind tests and 7 other molecular crystals to validate their procedure. Until now, the published blind tests included 26 different systems. Therefore, the authors should comment in the main text on why only these 8 particular systems were selected. The authors seem to focus only on single-component crystals involving relatively small and rigid molecules, which is of course reasonable for an initial validation of a new procedure. However, the performance for co-crystals, salts, hydrates, and crystals involving larger and flexible molecules cannot really be assessed yet. Therefore, the authors should also explicitly mention in the main text and the conclusion which types of molecular crystals were benchmarked.

In response to Reviewer 1, we have added in the abstract and in several other places that our current CPS(aiFF) method works only for rigid monomers. Thus, it should be now clear why only rigid monomers have been selected for our test set. We have also now explained why we believe that when CSP(aiFF) will be extended to flexible monomers (in a hybrid *ab initio*-empirical manner), it should lead to improvements over CSPs with empirical FFs. We have added now a statement in the Conclusion explaining why the CSP(aiFF) method should work equally well for cocrystals as it does for homogeneous crystals. Also, a statement on size of molecules was added.

2.) It might be beneficial for several readers to briefly also mention how much more expensive the newly developed force field is compared to the typical force fields used in CSP.

This comparison was already included in Fig. 3. The PACK+MIN bars are about twice higher for aiFF than for an empirical FF. We have now added explicit statements in the discussion of this figure.

3.) Please add some computational details to the description of the ZPVE and thermal effects calculations. Were they performed directly in VASP or with an external tool like phonopy? In case of finite differences, which supercells and q-grids were used?

The ZPVE and thermal effects were calculated within the harmonic approximation using Phonopy 2.8.1. We performed calculations at the Γ -point using a supercell of at least 10 Å length in each direction and an $8\times 8\times 8$ mesh. We have added this information at the end of Methods.

The authors have made some changes and responded to the earlier review. My responses to these responses are included below. My overall opinion of the work is unchanged. I think that this work is very nice, but it is mainly of interest for a more specialized audience.

If the work is published, some revision is necessary:

- i) The results with the W99 force field need checking. I ran calculations and find contradictory results to what they report.
- ii) The issue of “first principles” vs “empirical”, while not so important in my view, is problematic. The current work makes use of empirical force fields, unless I misunderstand the use of OPLS in a couple of places (in the selection of training points for the aiFF and in initial optimization of crystal structures in UPACK). This use should be more transparent if the authors choose to highlight similar use of force fields in other work.

More detailed responses

1. One problem is that a similar approach exists and has been published: reference 23 (Tailor-made force fields for crystal structure prediction) describes an approach to fit a force field to data derived solely from electronic structure calculations, whose validation is presented in ref 22. I do not know why the current manuscript describes these as “empirical FFs tuned for the system considered to DFT+D calculations for dimers of the molecules involved.” As far as the publications describe, these are force fields that are fitted solely to ab initio data. Note also that the methods described in ref 23 are applied to flexible molecules, unlike the rigid molecule validation that is presented in the current manuscript. Note that refs 22 and 23 are from 2008. This is not new methodology.

We believe the approach of Neumann et al. (Refs. 22 and 23) is significantly different from our approach, in fact, in our opinion it is not similar at all. We have extended the description of this method on page 2 to emphasize the differences. First, as we say in the manuscript, while the former approach is the most successful CSP method on the market, it cannot be considered as a fully first-principles method, i.e., we cannot agree with the reviewer that the tailor-made FFs of Neumann et al. are fit “to data derived solely from electronic structure calculations”, while our approach is of this type. The starting point for a tailor-made FF is the Dreiding empirical FF (see Ref. 23, p. 9812). The parameters of the Dreiding FF are, as stated several times in Ref. 23, “refined” using ab initio lattice energies and their derivatives (the use of the word “dimers” in our text was a mistake which has been fixed). In contrast, aiFF uses no experimental information and fits all parameters from scratch. FFs that are obtained by modifications of empirical FFs are often called “tuned empirical FFs” and this was the phrase used by us. We would like to emphasize that the fact that the tailor-made FFs of Neumann et al. are not obtained fully from first principles does not constitute any criticism of this method, but does point out to a significant difference between the two approaches. The second difference is that the ab initio data that are used to tune tailor-made FFs are obtained from pDFT+D calculations on crystals, while CSP(aiFF) uses SAPT calculations for dimers. While each approach has its advantages, the differences between them are significant. One may also add that to get the initial crystal structures for the pDFT+D calculations in the method of Neumann et al. an empirical FF has to be used. Furthermore, the pDFT+D method used by Neumann et al. does have an empirical flavor and in fact the title of the Neumann-Perrin paper in J. Phys. Chem. B 109, 15531 (2005) is: “Energy Ranking of Molecular Crystals Using Density Functional Theory Calculations and an Empirical van der Waals Correction”. Finally, some details of the Neumann et al. approach have

not been published, no source codes are available, and only commercial software realizing this approach exists. Although as the reviewer stated, this methodology was published in 2008, to our knowledge, there was no effort by other researchers to create an open-source version.

The fact that the method of Neumann et al. applies to flexible monomers is indeed an advantage over the aiFF approach presented in our manuscript. We discussed this restrictions and ways to extend the CSP(aiFF) method to flexible monomers in the section “Neglected effects” on page 5 (this text has now been expanded). We are currently working on such an extension, but we believe that crystals with rigid molecules are an important enough and large class of crystals to make our method relevant for the field. In the past, CSPs for such molecules were a laborious and daunting task, while we show that such CSPs can be easily performed with CSP(aiFF).

I agree that there are significant differences in the method described by Neumann et al and the present work. However, the description in the current manuscript: “This protocol uses a tailor-made FF which is obtained by refining parameters of an empirical FF to reproduce as close as possible pDFT+D lattice energies (and their derivatives)” is inaccurate. Reading through ref [23], there are several places where parameters from the Dreiding force field are used as starting points: the bond stretch parameters, angle bending parameters and they seem to take some bond increment parameters from Dreiding for the electrostatic model. Any optimisation of force field parameters requires a starting point and I don’t understand how the use of empirical force fields for starting points of a few parameters really distinguishes the approaches. Ref [23] also states that they use the same functional form as Dreiding for a hydrogen bond term and improper torsions. However, the intermolecular interactions (apart from a part of their electrostatics) do not seem to come from an empirical force field. The current paper is all about intermolecular interactions because molecules are treated as rigid. It seems that the method presented by Neumann et al, if applied to fully rigid molecules (as done here) would be very close to first principles. Furthermore, the authors say that they are extending their method to apply to flexible molecules by *combining it with an empirical intramolecular force field* (see below). So, to summarize, both methods are quite close to first principles when applied to rigid molecules, but both make use of empirical force fields in some way when they are applied to flexible molecules.

To me, the distinction between “empirical” and “first principles” is overplayed. Yes, Neumann and Perrin (J. Phys. Chem. B 109, 15531 (2005)) referred to their dispersion model as “empirical”, but here is their description in the paper: “Following Wu and Yang,⁹ we have fitted atomic C_6 coefficients to molecular C_6 coefficients derived from dipole oscillator strength distribution by Meath and co-workers.²⁸⁻³⁴ The method employed by Meath and co-workers is a complex mixture of theoretical and experimental techniques.” I’m not convinced of the importance of using absolutely no experimental information in determining a model. Why is this important, apart from when we want the model to work on atoms for which this experimental data is missing? However, this is not what is done in the current work. I could obtain the Neumann and Perrin C_6 coefficients for all 18 molecules studied here.

The authors also make the point that “One may also add that to get the initial crystal structures for the pDFT+D calculations in the method of Neumann et al. an empirical FF has to be used.” Why is this a problem? In fact, if I read references [37] and [38], which describe the method used in the current work, they use the OPLS force field in generating points at which they fit their energy model: “The first rejection criterion uses a guiding potential to weight the grid point distribution more

strongly in the more attractive regions of the potential. We use in the initial iteration of grid generation the OPLS-AA potential developed by Jorgensen and co-workers.” The Methods section of the current paper also describes the use of OPLS when using UPACK for CSP: *“We found that it is advantageous to perform Stage 4 first with the OPLS-AA FF, i.e., using the original UPACK path including clustering, and then minimize the reduced set using aiFF. The procedure was chosen not to save time, although it does result in minor savings, but to avoid minimizations ending up in “holes” of an FF, i.e., unphysical minima at very short intermonomer separations.”* So the OPLS force field seems fairly integral to the optimization procedure before aiFF is applied.

I don't think that this is a weakness. The empirical force fields – Dreiding and OPLS – exist, so make use of them. I am just making these points to show that it is hard (and perhaps pointless?) to try to completely remove the use of empirical force fields from the process. Also, that I see similar levels of use of empirical force fields in both works.

I strongly agree with the authors about the availability of the method (as I already stated in my original review). It is much better to have a method that is open source.

The details are different, such as the use of SAPT(DFT) for generating reference data. Also, it is welcome that the methods described here use open source software, making it more widely available, but this on its own does not lead to me think that the work is of a level required for Nature Communications. Also, given the results presented in ref 22, the statement “Such high level of predictivity is unprecedented for a complete first-principles CSP protocol” is not appropriate.

We hope that our arguments given above will convince the reviewer that CSP(aiFF) is a method fundamentally different from the method of Neumann et al. and only the former method is a fully first-principles one. If this is the case, the sentence quoted is actually accurate. Indeed, if the differences were only due to our use of SAPT and providing open-source codes, we would not have sent our manuscript to Nature Communications. However, the differences are substantial as described above. We believe the importance of our work stems from combining recent developments in ab initio methodologies and their numerical implementations [points (a)-(d) described on page 2 of our manuscript] into a reliable CSP protocol.

I think that we disagree on i) whether a completely first principles protocol is necessary, and ii) when do we choose to call a protocol first principles. Neumann and the current work have made use of Dreiding and OPLS in some very sensible places. Both make use of these force fields in how they would be applied to flexible molecules (although the application to flexible molecules is not part of the current paper).

2. Expanding on the first point, the validation here involves molecules that are too simple to make the claim that the method “will be a transformative change of the whole field of CSPs” (from the conclusions section).

As stated above, the class of rigid molecules is actually important on its own. We could have included larger molecules but since there are no reasons to doubt that the method will work for any size molecule that one can afford computationally (in fact, our group is now developing an aiFF for a 100-atom monomer), there was no point to do so.

In our opinion, the molecules included provide sufficient evidence that one can expect “transformative change”, i.e., the field of CSPs will transform from the use of empirical FFs to first-principle ones (we have clarified our use of the word “transformative” in the Conclusions). Although in the work presented in the manuscript we have not applied aiFF to crystals with flexible monomers, our results for the rigid ones indicate that an extended CSP(aiFF) method should work well for flexible monomers. The reason is that the flexible-monomer empirical FFs consist of uncoupled intermonomer and intramonomer parts. The former are, for a given geometry of a monomer, exactly the same as our rigid-monomer aiFFs. Thus, replacement of the intermonomer parts of empirical FFs by aiFFs should improve accuracy of such FFs

Yes, there are important classes of rigid molecules. However, a new method for rigid molecules will not transform the field of CSP. Basically, whether the field is transformed is speculation, and we don't agree. This is very nice work, but I do not think that it needs to be sold as “transformative”. I am also less confident that the extension to flexible molecules will be so straightforward: the approach (see below) is to couple the first principles intermolecular interactions with tuned versions of empirical force fields. These intramolecular force fields often fail, as has been found in the blind tests of crystal structure prediction and elsewhere.

Their introduction discusses the blind tests of crystal structure prediction, which have moved on from such rigid molecules. The method here assumes rigid molecules: their geometries are held fixed at the optimized geometries of the isolated monomers. While the discussion states that the method can be extended to flexible molecules, this is not demonstrated. I am less certain that the extension to flexible molecules will be as straightforward as the authors claim.

We are currently working on such an extension and the results are encouraging. As stated in the manuscript on page 5, we assume that the intermolecular and intramolecular degrees of freedom are uncoupled, as is the case for all empirical FFs and also for the tailor-made FFs of Neumann et al. Since the analytic form of our aiFF depends only on atom-atom distances, the interaction energy can be computed for any deformations of monomers. We call an aiFF developed for rigid monomers and used for different monomers geometries a “flexibilized” aiFF. This intermonomer FF is used together with the intramonomer term of an empirical FF regularized to have the minimum at the ab initio equilibrium geometry. We apply UPACK to perform flexible-monomer CSPs with such a hybrid FF. Since the quality of predictions of flexibilized aiFFs decreases with increased deformation of monomers, this quality is monitored by performing SAPT(DFT) calculations for nearest neighbour dimers in polymorphs predicted by UPACK. If the difference between interaction energies from aiFF and from SAPT(DFT) becomes larger than some assumed threshold, a new aiFF is developed using deformed monomers found by UPACK. This is less expensive than the development of the original aiFF since the previously computed grid points can be reused. A condensed version of these explanations was added on page 5.

Inter- and intramolecular terms in force fields are not truly uncoupled. Force fields always have an awkward way of dealing with non-bonded atoms within the molecule: 1-2 and 1-3 interactions are taken care of by bond stretch and angle bend terms, anything further away than 1-4 is treated in the same way as intermolecular interactions, while 1-4 interactions involve a combination of dihedral terms and (sometimes scaled) non-bonded terms. If the parameters are not developed together, this

is not expected to work in a general sense. I cannot judge the statement at the end of page 5: “One can expect that such a replacement should lead to improved predictions in flexible-monomer CSPs” without the authors actually presenting results from this approach.

3. The results, before re-optimization of structures using pDFT-D, are not very different from what can be achieved with good quality empirical force fields. Comparing the rankings of the observed crystal structures (“The CSP(aiFF) protocol ranked the experimental polymorph as number 1 in 5 cases, as number 2-6 in 7 cases, and as numbers 9, 9, and 16.”) to validation studies with empirical force fields that involve high quality electrostatic models (see, for example, *Crystal Growth & Design* 2005, 5, 3, 1023–1033, <https://doi.org/10.1021/cg049651n>) show similar distributions of rankings. The study presented in *Crystal Growth & Design* 2005, 5, 3, 1023–1033 looked at a larger set of molecules of similar size to those studied here and found that approximately a third of observed crystal structures were ranked as #1 in CSP and two thirds of observed structures in the top 5 ranked structures. This is similar to what has been presented here. There might be a modest improvement, but I am not sure that this would be significant based on 15 molecules. The important advantage of the current approach is that it can be applied equally well to molecules with atom types or functional groups that are not well represented in empirically derived force fields (it is unfortunate that the molecules studied here do not contain any such examples and would all be modelled quite well with empirical force fields). However, for an empirical force field, this can be addressed by a one-off extension of the parametrization.

We cannot agree with Reviewer’s 1 statement that the quality of CSPs with empirical force fields “is similar to what has been presented here”. The data from the paper by Day et al. in *CG&D* 2005, quoted by the reviewer, should not be compared with our rankings, in fact, they are not rankings. Our rankings are obtained in blind-test fashion, i.e., using no experimental information. Then the purely geometrical RMSD20 criterion is used to find which predicted polymorphs are close to the experimental ones. Day et al. did not perform the second of these steps.

I do not understand this comment. The paper (*CG&D* 2005) discusses rankings of the experimentally observed crystal structures “As in our previous study [*Crystal Growth & Design* 2004, 4, 6, 1327–1340], we discuss the results in terms of a relative energy, ΔE , and the number of predicted structures that are lower in energy than the observed crystal structure, N_{lower} .” This previous study describes the development of the CCDC’s crystal structure comparison tool for comparing structures and calculating RMSDs and states “This method was also used to locate the position of the observed structure within the predicted list.”

Instead, they compare their results to “the experimentally observed structures after energy minimization with the W99 +multipoles model potential”.

This text seems to be taken from the description of a figure in the paper, where the energy of the experimentally determined structure is shown in comparison to the results of CSP. I don’t think that this means that this is how they made all of their comparisons.

Day et al. did not report what is the rank of the experimental polymorphs among theoretically predicted polymorphs. The quantity N_{lower} given in their Table 1 is just the number of theoretical polymorphs with the energies below the experimental polymorph. These numbers are not necessarily indicative of a priori rankings and N_{lower} should not be compared with our rankings.

I don't understand the distinction. How is N_{lower} not indicative of a priori rankings when it is calculated in the same way that rankings are calculated in the present paper (by comparing predicted structures to the experimental structure)?

The comment of Reviewer 1 prompted us to check how good are the predictions of the empirical FF (W99+point-charges model) used by Day et al. compared to aiFFs. We obtained the point charges using the CHELPG method [Breneman and Wiberg J. Com. Chem. 11, 361 (1990)]. The results listed in Table 1 below demonstrate that the performance of aiFFs is significantly better, in fact, qualitatively better. For example, aiFF has 94% of cases ranked at positions 10 or better, while W99+charges only 33%. This is indeed a qualitative difference for technological applications. These results are described in a new section "Performance of an empirical FF" on page 4 and in SI. The work of Day et al. has shown that an electrostatic model using atomic multipoles performs better than charges-only model, but the differences between the two models are much smaller than the difference between W99+charges and aiFF, as shown in Table 2 below.

4

Table 1: Comparison of ranking of the 18 polymorphs by aiFF and by W99+charges FF. The numbers give the percent of the polymorphs at a given range of ranks, with cumulative values in parentheses. MS denotes missing parameters in W99 FF. The column >100 counts CSPs that did not include the experimental polymorph within 3112, 568, 1272, and 2463 lowest lattice energy polymorphs for monomers I, II, benzene, and deferiprone, respectively.

	1	2-10	11-20	21-100	>100	MS
aiFF	28	67 (94)	6 (100)			
W99+charges	6	28 (33)	11 (44)	22 (67)	22 (89)	11 (100)

Table 2: Comparison of charges-only and multipole moments models from Table 1 in Day et al. The percentages are computed using all 65 rows in this table.

N_{lower}	1	≤ 5	≤ 10	≤ 20
charges	29	53	69	81
multipoles	34	65	82	89

Reviewer 1 wrote that it is unfortunate that our set of molecules does not include atom types or functional groups that are not well represented in empirically derived force fields. Our results with the W99+charges show that actually there are several such examples. For two molecules the CSPs could not be done since the atom types needed are not available in W99+charges. CSPs based on W99+charges missed 4 out of 18 polymorphs completely (these polymorphs were not present on the list of polymorphs that were examined). Clearly, some functional groups in these molecules are poorly represented by the W99+charges FF. One may add that molecules like DNBT and TNB are not well parametrized in most empirical force fields. The W99+charges FF actually performs for these crystals better than other empirical FFs since the training set of W99 included a large number of nitrogen-containing molecules. Still, the RMSD20 for DNBT is 0.81°A , slightly above the CCDC threshold of 0.8°A .

I was surprised by these results, since Day et al report that both benzene polymorphs are found in CSP studies, and also both polymorphs of molecule I (CSD reference code XULDUD and XULDUD01). I ran crystal structure prediction calculations using the W99 force field and atomic charges (fitted to the molecular electrostatic potential of a PBE0/6-311G** electron density). The search method is not UPACK, but this is not the important issue. Both polymorphs of benzene are located. Here is a screen capture of a match found in the benzene set to the Pbc_a polymorph:

I also ran calculations on molecule I and find both polymorphs, in contrast to what the authors report. The RMSD for the P21/c polymorph of I is rather high, but it is a 20/20 match in Mercury and I believe that it is close enough that re-optimization with periodic DFT would optimize the structure to a much better match (ie. strategy 2 in the paper). I have not run calculations on the other two molecules for which failure of CSP with W99 is reported here, but the results for benzene and molecule I mean that I am not confident in the searches that have been performed. I suspect that the sampling applied using UPACK is insufficient and that this is not a force field issue, or we have a difference in how W99 has been implemented. It seems odd that they were found with the OPLS + aiFF method, though.

4. The cost analysis (centered on Fig 3) seems unfair when compared to existing methods. The alternative strategies are set up to be more expensive. For example, “Strategy 2 differs from Strategy 1 by the use of an empirical FF in the PACK+MIN stage and by performing pDFT+D calculations for 100 polymorphs with reoptimization of geometries, a strategy similar to that used in Ref. 41.” The complexity of the systems studied in ref 41 is much higher than those studied here, so it is natural that more crystal structures must be considered in the final stages of optimization. This has to do with the complexity of the system rather than the method: molecule XXII from the blind tests is common to both studies (the current manuscript and ref 41) and did not require so many structures in ref 41.

First, Strategies 2 and 3 were not intended to criticize existing methods. We mentioned such methods just to show that similar approaches have been used. We have now removed these mentions and stated that these strategies are hypothetical. With the new results obtained using the W99+charges FF, we were able to evaluate these strategies in more precise terms. In particular, we can now state that the success rate of Strategy 2 is 67%, see Table 1 here. We also have now realized that the comparison with the work of Hoja et al., Ref. 41, was inappropriate, but for reasons different than given by the reviewer: the 100 polymorphs were obtained in that work not from an empirical FF, but using the tailor-made FFs of Neumann et al., which are, of course, much more predictive than any empirical FF. This is the reason why, as the reviewer rightly points out to, the 100 polymorphs were sufficient for crystals with flexible monomers. BTW, we are not sure why the reviewer implies that Hoja et al. used less than 100 structures for system XXII. What they say is: “As

the foundation for the presented stability ranking approach, we use the top 100 molecular crystal structures (for every system of the latest blind test) from the above mentioned sampling approach of Neumann and co-workers using GRACE.”

The section on the timing comparison is better. The strategy where all calculations are performed with solid state DFT is still unrealistic. Yes, one participant did this for one molecule in a blind test. It is not something anyone else would do. Personally, I would remove Strategy 3. Why make the comparison if it is purely hypothetical?

Although the section is improved, I still believe that the authors are making unfair comparison. They now quote strategy 2 as leading to a 67% success rate. Firstly, this is based on 18 structures. This is a small number, so the error bar on this success rate is large. This 67% is 12 successes out of 18. The 6 failures include 2 systems for which calculations were not performed because the authors chose an empirical force field without the required parameters (for F, Cl, Br). The authors could have chosen a different force field that has the required parameters. For example, Price and co-workers list parameters for all but Br that have been used with the FIT force field:

<https://doi.org/10.1039/C004164E> Other force fields exist that also include Br. So it is artificial to include these as failures. Also, I am now unsure about the other 4 failures. As mentioned above, I ran benzene and molecule I and find the polymorphs that are reported here not to be found.

Regarding the 100 structures: Yes, Hoja et al re-optimised 100 structures of each molecule in ref [43]. The point is that this was not required for the small, rigid molecule XXII, where the force field already ranked the experimental structure as #2. Presumably, 100 structures were included for all molecules for consistency. This was not required for the small, rigid molecule. If Hoja's work had only looked at rigid molecules, I suspect that they would have used fewer than 100 structures.

Overall, the study presents a useful method for CSP of organic molecules. It is clearly a useful method that performs well on some fairly simple test systems. However, it is not the level of advance that the authors claim and the comparison to existing methods is not fair in many places. I suggest that the work is more appropriate for a more specialized physical chemistry or materials chemistry journal and that the authors are more careful with their comparison to existing approaches. I would also like to see validation on systems that highlight the strength of the method more strongly: molecules with atoms that are not well-represented by empirical force fields and molecules with intramolecular flexibility, where force fields also have important limitations in their accuracy.

We will first respond to the statements of Reviewer 1 from the opening paragraphs: “However, I think that the authors have exaggerated the [...] the improvement that [their method] offers over existing methods” and the statement made above “it is not the level of advance that the authors claim”. The level of advance of our work is determined by our combination of a reliable and efficient ab initio method with a method for automatic fitting of ab initio interaction energies and then the use of the resulting aiFFs (of a form more elaborate than that of most empirical FFs) in CSPs, followed by reranking with pDFT+D. We believe this is a true advance and no similar protocol has been available, therefore, our work does constitute a significant improvement over existing methods in terms of theory development. In particular, we hope we have convinced the reviewer that our method is not similar to that of Neumann et al. An important difference between the two approaches is that the latter is partly based on empirical input, while our is a completely first-principles method. We believe that moving from empiricism to fully first-principles treatments is an

important advancement in any field of science. The next criticism is that the comparison to existing methods is not fair in many places. One of those comparisons was the relation between the Neumann et al. and our method, already discussed. Another comparison concerned the improvements of predictability of the pure aiFF approach to that of empirical FFs, which the reviewer considers to be only “modest” based on comparisons to Day et al.. We first point out that the quantities that the reviewer extracted from the Day et al. paper are not rankings of a blind search. We then performed CSPs on our set of molecules applying the W99+charges FF used in Day et al. The results show that the improvement due to the use of aiFF is not modest, but quite dramatic, really moving CSPs to a new level of quality of predictions. This comparison also shows that we have not exaggerated the improvement CSP(aiFF) offer over existing methods. Finally, the reviewer believes our costs comparisons are unfair. This is a misunderstanding, we had no intention to compare costs to existing methods. It is now clearly stated in the manuscript that the methods considered in costs comparisons are hypothetical. The criticism that we do not provide examples how our method works on “molecules with atoms that are not well-represented by empirical force fields” has been answered by our application of the W99+charges empirical FF to our set of molecules: W99+charges fails really badly on several of them.

We disagree on quite a lot here, but this has been covered. I am also not in agreement with “We believe that moving from empiricism to fully first-principles treatments is an important advancement in any field of science.” I admit that there is an elegance to being able to do things with no empirical element. However: i) there are empirical elements to the current work – the OPLS force field comes into things in a few places and ii) I am happy with an empirically-based method, as long as it works.

We cannot provide examples of how CSP(aiFF) works for “molecules with intramolecular flexibility” since the current version of our CSP protocol works only for rigid monomers (aiFFs have been applied in molecular dynamics simulations with flexible monomers, for example in the 6th blind test, but this protocol is different from what we propose here). The extension to flexible monomers is under development in our group. However, as we argued above, the class of rigid monomers is, in our opinion, broad enough for our method to be of significant importance. Moreover, as we also discussed above, there are good reasons to believe that the use of aiFFs in CSPs with flexible monomers will also result in improved predictions.

Finally, let us address the criticism from the opening paragraphs that we “have exaggerated the impact that this work will have”. We do believe the impact will be significant for several reasons. First, we have developed the first fully first-principles CSP method which can be used in an automated fashion. Second, CSP(aiFF) performs dramatically better than similar approaches based on empirical FFs. Moving from empiricism to first principle methods is an important advancement in any field of science. Third, although the method works now only for rigid monomers, it is evident that its extension should improve flexible-monomer CSPs. While the reviewer criticized our statement that “the proposed CSP protocol should result in a transformative change of the field of CSPs”, we believe the impact of our work will be that researchers working on CSPs will start using aiFFs instead of empirical FFs. Such a transformation has already happened in the field of van der Waals clusters and in molecular dynamics simulation of fluids of small molecules. This transformation will be helped by the fact that the CSP(aiFF) protocol is an easy to use (since it is highly automated) approach freely available to all researchers. Until now, there has not been any method on the market with all these attributes.

Most of this is already covered in the responses above. I congratulate the authors on this nice work. I am only trying to make a judgement on the impact and the comparison to previous existing methods. I believe that the work is important, but mainly of interest to a fairly specialized audience.

Reviewer #2 (Remarks to the Author):

The authors have satisfied my concerns in the revised manuscript and I support publication.

We really appreciate the further comments by Reviewer 1 and in particular the fact that the reviewer spent time to check some of our calculations. However, most of the current comments concern differences between reviewer’s and our views on the importance of various approaches in the research field considered, in particular concerning the importance of *ab initio* versus empirical approaches. In our opinion, even if the reviewer does not appreciate the former approaches, he/she should appreciate that making them work in CSPs is an important progress in the field.

Below we reply to all the comments of Reviewer 1. However, since adding one more layer to the 9-page long exchange would make such document very hard to read, we do not copy the previous text but refer to it.

- (a) The reviewer performed CSPs [point i) on page 1 and page 7 of the report] with the W99+q FF for the benzene and system I crystals and found two polymorphs in each case, while we found only one in each case. It turns out that omitting the *Pbca* polymorph of benzene was our mistake, but including it is inconsequential for the comparison. The *P2₁/c* polymorph of system I found by the reviewer should not be included on the list, as explained below.

After reanalyzing our results, the *Pbca* polymorph of benzene has been found at rank 31 with $\text{RMSD}_{20} = 0.27 \text{ \AA}$. Our mistake was that we searched for an experimental polymorph only in its space group. We have now searched all default space groups for all crystals, and the second polymorph of benzene was found in the *P2₁2₁2₁* space group. It was then transformed into *Pbca* using PLATON. No other missing polymorphs have been found. We have fixed this mistake in the text and in the SI. We thank the reviewer for checking our work.

On the other hand, we were not able to find the *P2₁/c* polymorph of system I within the $\text{RMSD}_{20} = 0.8 \text{ \AA}$ limit set by CCDC on our list of 3112 polymorphs generated by UPACK. We were able to find this polymorph only by significantly increasing the tolerance: the polymorph is ranked at 130 with $\text{RMSD}_{20} = 1.20 \text{ \AA}$. In our opinion, this RMSD_{20} is too high to qualify this structure as the experimental crystal (also the overlay of the two structures looks really bad). Thus, we believe our reported results for this polymorph are right. Note that the RMSD_{20} threshold is not used in aiFF@CSP until the search for experimental crystal is performed. If W99+q were to be used instead of aiFF, the *P2₁/c* polymorph of system I would have been rejected based on the lattice energy ranking.

- (b) The reviewer stated [point ii) on page 1, pages 2-3, page 9] that our approach is not fully *ab initio* since it uses the empirical OPLS FF at two stages. This is true, however, the important point is that this use has no effect on the final results and that both uses of OPLS could easily be avoided.

In the development of aiFFs with the autoPES software, the OPLS FF is used to generate grid points in the first iteration of the fit. However, autoPES actually has an option of generating the initial grid at random. The final fits generated with both choices are essentially identical, but the second option requires more iterations. Note also that the asymptotic parameters (vdW constants and charges) are always obtained fully *ab initio* without using any empirical information.

In CSPs with UPACK, we do use OPLS in early stages of lattice-energy minimizations, but in those with MOLPAK, we do not use any empirical FF. When both methods are applied to the same crystal, the results are essentially the same. Furthermore, we could have fitted a 12-6-1 FF form to our *ab initio* data points and use it instead of OPLS in UPACK. A similar statement has been added in Methods.

We could redo our calculations avoiding any use of OPLS, but it would be a waste of time as it should be clear already that this use is inconsequential for the aiFF@CSP method.

- (c) The reviewer believes [point ii) on page 1 and page 2] that since both aiFF@CSP and the Neumann *et al.* method use empirical information, both methods are in the same ballpark in terms of their *ab initio* character. In our opinion, the differences in the use of empirical information are essential. As stated in (b), our method can easily avoid the use of any empirical information. We believe that the same is not possible for the method of Neumann *et al.* Let us suppose that the initial polymorphs are generated in this method randomly. Since only a few dozens of them can be selected for pDFT+D calculations, the initial optimization of FF parameters would be made on polymorphs very different from those experimentally relevant. Also if all initial values of parameters were to be random (within some ranges), a generation of an FF would probably be almost impossible in this approach.

On page 2, Reviewer 1 states that the intermolecular 12-6 starting parameters in the Neumann *et al.* method do not come from the Dreiding FF. We were not able to find this in the papers by these authors. We believe if that were the case, it would have been stated explicitly.

We agree that the origin of the dispersion function parameters in the pDFT+D method applied by Neumann *et al.* is a marginal issue (we used the word “empirical flavor”).

While we stress once more that the Neumann *et al.* approach is currently the most reliable CSP method, we would like to give an example on why truly first-principles approaches are important. In 1976, Matsuoka, Clementi, and Yoshimine (MCY) published (JCP 64, 1351 '76) an *ab initio* potential for water, which predicted properties of liquid water very well. For the next 20 or so years, none of the *ab initio* potentials (including some developed by Clementi’s group) could approach the accuracy of MCY predictions despite using higher levels of theory and more grid points. It later turned out that the MCY parameters were obtained starting from an empirical potential fitted to properties of liquid water, which were then “refined” to reproduce the computed interaction energies.

- (d) In more fundamental terms, we disagree with the reviewer that the issue of first principles versus empirical approaches is “not so important” [point ii) on page 1, also page 9]. In our opinion, all the development of science shows this importance (think of Kepler and Newton).
- (e) On page 4, the reviewer questions again our use of the word “transformative”. We do believe our work is transformative in the sense specified near the end of Conclusions.

Therefore, we kept the sentence in the Conclusion, but revised the last sentence of the abstract removing the word “transformative” from there.

Obviously, when one will use aiFFs together with empirical intramonomer FFs, this approach will not be anymore a fully first-principles one, but it will remain such in the intermonomer sector of FFs. BTW, we are also working on refining intramonomer FFs using *ab initio* information.

A minor point: the reviewer wrote that “intramolecular force fields often fail”. In fact, the intramonomer components of universal FFs are generally considered more reliable than the intermonomer ones.

- (f) On pages 4-5, the reviewer points out that the intermonomer components are used in empirical FFs in the intramonomer sector and calls it a coupling of the two components. In our use of the word “coupled”, we meant something else: in coupled FFs, the intermonomer parameters have to depend on geometries of monomers.

We actually do not plan to use aiFF parameters in intramonomer components when we move to flexible molecules. Thus, the type of coupling that the reviewer described will not be broken: the intramonomer FF will be the same as in simulations with empirical FFs, while the intermonomer component will be significantly improved. We believe that based on these premises, one can expect improved predictions for flexible monomers.

- (g) On page 5, the reviewer raises again the question of what Day *et al.* did in their 2005 CG&D paper. We admit that their description is not clear, but the text quoted by the reviewer supports, in our opinion, the view that the experimental structures were not found using a blind test type of protocol. The phrase that we quoted is on page 1024, Sec. 3, lines 13-15. In any case, since we applied W99+q to our set of crystals, there is no need to discuss the Day *et al.* paper anymore. Also the fact that aiFFs give such dramatically better predictions than W99+q on our set of crystals clearly indicates that it is not possible for these two FFs to give similar quality predictions on Day’s *et al.* set of crystals.

- (h) The reviewer questions on page 8 the inclusion of Strategy 3. We believe it is important to include this strategy since it is a strategy that assures 100% success rate with most empirical FFs. We do discuss the possibility that one can select any number of polymorphs between 100 and 25,000 to be used in this strategy.

Concerning Strategy 2 (in which case, after fixing the benzene mistake, the value of 67% changes to 72%), we do realize that there are empirical force fields that contain F, Cl, Br, but it was the reviewer who suggested comparisons with W99+q. The readers can easily make their own judgement on this issue. Also, one should add that aiFF@CSP achieves 100% success rate with rank up to 16, not 100. The main point raised by the reviewer in his/her previous report was that our aiFFs are not that much more predictive than empirical FFs. Haven’t we proved beyond doubt that they are? Why to raise then such minor points as missing atoms?

There are now no references to Hoja *et al.* paper in this fragment, so we do not need to discuss what they they did and what they could have done.

- (i) On page 9, the reviewer expresses an opinion that our work is “mainly of interest to a fairly specialized audience”. Depending on the assumed range of the word “fairly”, this is true for any given fraction of papers published in Nature Communications.